# Harnessing Stratospheric Diffusion Barriers for Enhanced Climate Geoengineering

Nikolas O. Aksamit[1], Ben Kravitz[2,3], Douglas G. MacMartin[4], George Haller[1]

[1]Institute for Mechanical Systems, Swiss Federal Institute of Technology (ETH), Zürich, Switzerland.
[2]Department of Earth and Atmospheric Sciences, Indiana University, Bloomington, IN, USA.
[3]Atmospheric Sciences and Global Change Division, Pacific Northwest National Laboratory, Richland, WA, USA.
[4]Sibley School of Mechanical and Aerospace Engineering, Cornell University, Ithaca, NY, USA.

*Correspondence to*: Nikolas O. Aksamit (naksamit@ethz.ch)

**Abstract.** Stratospheric sulfate-aerosol geoengineering is a proposed method to temporarily intervene in the climate system to increase reflectance of shortwave radiation and reduce mean global temperature. In previous climate modeling studies, choosing injection locations for geoengineering aerosols has thus far only utilized average dynamics of stratospheric wind fields instead of accounting for the essential role of time-varying material transport barriers in turbulent atmospheric flows. Here we conduct the first analysis of sulfate aerosol dispersion in the stratosphere comparing a now-standard fixed-injection scheme with time-varying injection locations that harness short-term stratospheric diffusion barriers. We show how diffusive transport barriers can quickly be identified and we provide an automated injection location selection algorithm using short forecast and reanalysis data. Within the first seven days of transport, the dynamics-based approach is able to produce particle distributions with greater global coverage than fixed-site methods with fewer injections. Additionally, this enhanced dispersion slows aerosol microphysical growth, and can reduce the effective radii of aerosols up to 200-300 days after injection. While the long-term dynamics of aerosol dispersion are accurately predicted by short forecast transport barriers, the long-term influence on radiative forcing is more difficult to predict and warrants deeper investigation. Statistically significant changes in radiative forcing at timescales beyond the forecasting window showed mixed results, potentially increasing or decreasing forcing after one year when compared to fixed injections. We conclude that future feasibility studies of geoengineering should consider the cooling benefits possible by strategically injecting sulfate aerosols at optimized time-varying locations. Our method of utilizing time-varying attracting and repelling structures shows great promise for identifying optimal dispersion locations and radiative forcing impacts can be improved by considering additional meteorological variables.

## 1 Introduction

Stratospheric sulfate-aerosol geoengineering relies on triggering an atmospheric perturbation through deliberate injections of sulfate aerosols or their precursors (often $SO_2$) into the lower stratosphere to mimic the cooling effects seen after large volcanic eruptions [The Royal Society, 2009]. Over the last several decades, this has been suggested as a possible means of reducing some of the impacts of climate change [e.g., Crutzen, 2006]. There are, however, many open questions about the effects of

radiative forcing from sulfate injections [Kravitz and MacMartin, 2020]. The importance of choosing the altitude and latitudes of injection, and distribution of injection rates across those, has been clearly demonstrated, as well as adjusting injection locations based on the season [Visioni et al., 2020]. Additionally, even for sulfate aerosols, the method of dispersal will affect

aerosol size distribution, and hence the amount of material that needs to be injected. To date, many of these uncertainties are based on a climate response from fixed-injection locations [e.g. Robock et al., 2008; Heckendorn et al., 2009; Tilmes et al., 2017], a significant limitation for predicting dispersion in fully turbulent fluid flows. In fact, none of these studies consider the short-term variations of stratospheric winds or the organizing role of turbulent coherent structures in these time-varying flows. Driscoll et al. [2012], showed that it is impossible to correctly capture the impact of abrupt atmospheric perturbations on

surface climate without a well-resolved stratospheric model. With the great significance of stratospheric dynamics for teleconnections and the state of the atmosphere [e.g. Jaiser et al., 2013, Domeisen et al., 2018], how can we optimize where to put aerosols or precursors so that we have greater influence on the mean climate, and with better efficiency?

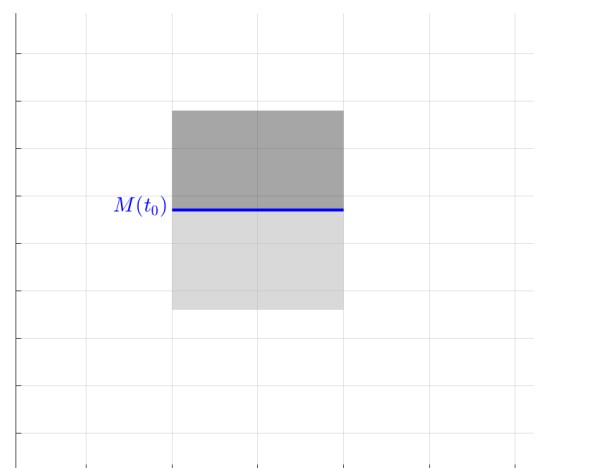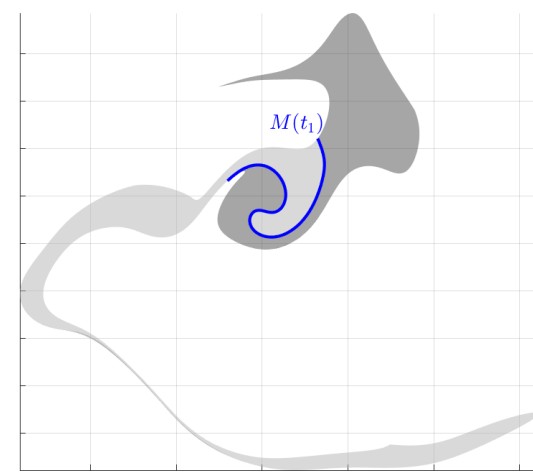

**Figure 1: Example of fluid particle advection for an unsteady geophysical 2D fluid flow from time $t_0$ to time $t_1$. For any arbitrary line of initial fluid particle positions, such as $M(t_0)$, that line will be a barrier to advective transport and**
**mixing. This is seen in the second panel as no dark grey fluid has crossed $M(t_1)$ to mix with the light grey fluid.**

While benchmark studies have been quite successful at understanding the mean climatic response of geoengineering in sophisticated Earth System Models [e.g. Kashimura et al., 2017; Kravitz et al., 2017], the injection protocols have all neglected presently-available short-term predictive information useful for optimizing particle dispersion. An efficient dispersion of

aerosol precursors is of crucial importance for aerosol coagulation [Kravitz and MacMartin, 2020]: the particle size distribution is a critical and sensitive parameter for accurately determining surface cooling, stratospheric warming, and changes in stratospheric dynamics [e.g., Rasch et al., 2008; Heckendorn et al., 2009; Tilmes et al., 2008; Niemeier et al., 2011]. By only considering average flow behavior, one limits geoengineering evaluations to simple injection protocols that do not fully exploit turbulence, coherence, and mixing in the stratosphere. This increases the likelihood of a heterogeneous spatial coverage

and localized high concentrations of aerosols, leading to enhanced coagulation and sedimentation rates [e.g., Pierce et al., 2010]. Without a more precise optimization of injection locations, we limit our ability to accurately model the full potential impacts of geoengineering.

Instead of standard fixed-locations, we propose a time-varying injection location protocol based on identification and prediction of short-term Lagrangian stratospheric transport barriers. This method harnesses the theory of Lagrangian coherent

structures (LCSs), a tool for highlighting the most influential material surfaces solely from fluid velocity fields without any further modeling of scalar transport [Haller, 2015]. For a given unsteady fluid flow, any arbitrary surface of fluid particles, $M$, will block advective transport across that surface over time as the surface deforms with the flow. This is shown in a real 2D velocity field of geostrophic ocean surface currents in Figure 1. Here, the blue line $M$ separates regions of light and dark grey fluid particles. As the fluid flows from time $t_0$ to $t_1$, $M$ is an advective transport barrier in that no dark grey fluid crosses $M$ to

mix with the light grey fluid. This result follows immediately from the continuity of the equations defining fluid motion.

Instead of looking for material barriers to advective transport, of which there are infinitely many, LCS theory identifies only exceptional distinguished material surfaces, such as those that are mathematically defined to be rotationally coherent, undergo minimal stretching over time, or locally attract or repel nearby fluid particles at a significant rate. One example of the latter two structures, termed hyperbolic LCS, and their time evolution in the same unsteady ocean flow are shown in Figure 2. Over

the time period $t_0$ to $t_2$, $M_A$ is the structure that is mathematically-defined to most effectively attract nearby particles, and $M_R$ repels nearby particles. By identifying exceptional material barriers, such as the saddle feature in Figure 2, LCS theory allows organization of turbulent fluid flows into coherent patterns in a mathematically-rigorous (non-empirical), physical and frame-independent manner [Haller, 2015].

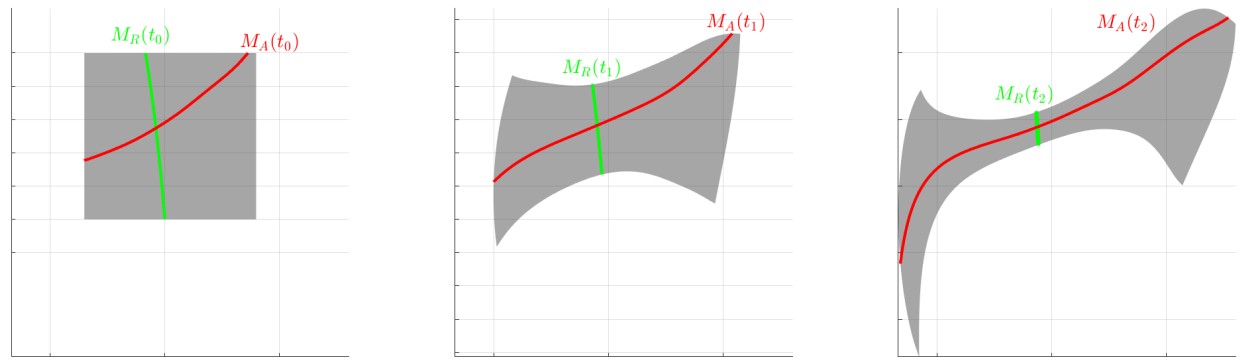

**Figure 2: Example of time evolution of fluid particles surrounding hyperbolic LCS in a geophysical fluid flow from time $t_0$ to $t_2$. $M_A$ is an attracting LCS (unstable-manifold) and $M_R$ is a repelling LCS (stable-manifold).**

Though using the mathematical definition of LCS to define atmospheric flow structures is quite restrictive, LCS have actually been identified throughout the atmosphere [Tang et al., 2010; Tallapragada et al., 2011; Rutherford et al., 2012;

BozorgMagham and Ross, 2015; Knutson et al., 2015; Wang et al., 2017].  Of particular relevance to the present research is the LCS work of Beron-Vera et al. [2012] who demonstrated how zonal jets behave as meridional transport barriers at high latitudes. Olascoaga et al. [2012] analyzed LCS in stratospheric winds to provide a rigorous definition of the transport barriers contributing to the loss of ozone from the Arctic ozone layer, and there was recent success in delineating LCS along atmospheric rivers [Garaboa-Paz et al., 2015]. Jupiter's Great Red Spot and zonal jets were identified as material transport

barriers through video analysis and LCS theory [Hadjighasem and Haller 2016]. Using a null-geodesic identification scheme, the northern polar vortex, a significant structure in high-latitude atmospheric mixing, was accurately identified as a transport-blocking LCS [Serra et al., 2017].  Lastly, Wang et al. [2017] were able to use a related diagnostic strain-tensor field to predict the location of space shuttle contaminant plumes in the thermosphere after 48 hours of transport. These previous results indicate the potential for the most influential LCS to be harnessed for geoengineering purposes. Specifically, hyperbolic LCS that

maximize or minimize dispersion may be used as time-varying injection locations that reduce coagulation of aerosols and increase their lifespan and utility.

Recently, Haller et al. [2018, 2020] derived an additional objective criterion that specifically identifies the strongest barriers and enhancers of diffusive particle transport. That is, one can identify the time-varying locations of material barriers in a fluid flow that maximize or minimize the diffusive contribution in the advection-diffusion equations over a given timeframe. They

have obtained a diffusion barrier strength (DBS) field whose ridges highlight the strongest diffusive transport barriers in forward-time fluid flow analysis and strongest diffusive transport enhancers by running a backward-time fluid flow analysis. Neither of these simulations actually require modeling the evolution of a diffusive scalar field, but still rigorously define the structures that are most influential to diffusive transport. For atmospheric science, this significantly reduces the computational burden for predicting how scalar fields will evolve as it provides quantitative information about future attraction and dispersion

patterns without needing complex numerical machinery to model the advection-diffusion equations, or making assumptions about their unknown initial- and boundary conditions. In comparison, the effective-diffusivity approach of Nakamura [2008] provides an a-posteriori visualization of Eulerian barriers, but only after scalar transport simulations have been performed. DBS fields, however, give an a priori (predictive) characterization of material barriers to diffusion without ever running diffusive simulations. This new technique increases the rigor of Lagrangian atmospheric analysis and removes ambiguity

arising from the lack of a universal definition of coherence in atmospheric LCS work. As such, the DBS field is perfectly suited for optimizing aerosol dispersion and is computable solely from available wind field forecasts and hindcasts or reanalysis.

In this manuscript, we evaluate simulated stratospheric flows with the aim of identifying diffusive transport barriers and informing injection site selection for enhanced stratospheric geoengineering via aerosols.  In doing so, we provide an initial

demonstration of the benefits of incorporating short-term atmospheric dynamics into geoengineering analyses and provide suggestions to better assess its potential impacts.  Our choice of dynamics-informed injections is evaluated against fixed-injection protocols via long-term metrics of pure advective transport, and for geoengineering scenarios simulated in a fully coupled climate-model. We find significant improvement in the ability of injected aerosols to both quickly surround the earth,

and to be able to achieve similar coverage with fewer injection sites. We then introduce further practical and logistical restrictions on the DBS-based protocol, and maintain our method's improved performance.

## 2 Methods

### 2.1 Climate Model Data

We use CESM2 (WACCM6) [Gettelman et al., 2019] under an SSP5-8.5 scenario to generate global wind fields at 72 levels for 18.75 years of simulation (Table 1). These fields were computed at a spatial resolution of 0.94° latitude and 1.25° longitude, with instantaneous output at 6-hour frequency. As vertical motion is minimized over short timescales along isentropic surfaces, and similar analysis has reliably identified transport barriers along these surfaces [Serra et al., 2017], we extracted wind fields on isentropes ranging from T=280 K to 1000 K with 20 K resolution. This is expected to provide a computationally efficient 2D analysis of material barriers to aerosol and tracer transport. We primarily focus on the T=540 K isentrope in the lower stratosphere (approximately 20-25 km ASL in the tropics) as these elevations are at the upper limit of currently practical aerosol injection heights. The DBS-injection protocols described herein rely only on 14-day windows of wind velocity and can be applied to wind data at any height. It is reasonable to assume that applying these methods elsewhere and optimizing injection locations to maximize dispersion at other heights would be beneficial for aerosol global coverage and similar results may be possible.

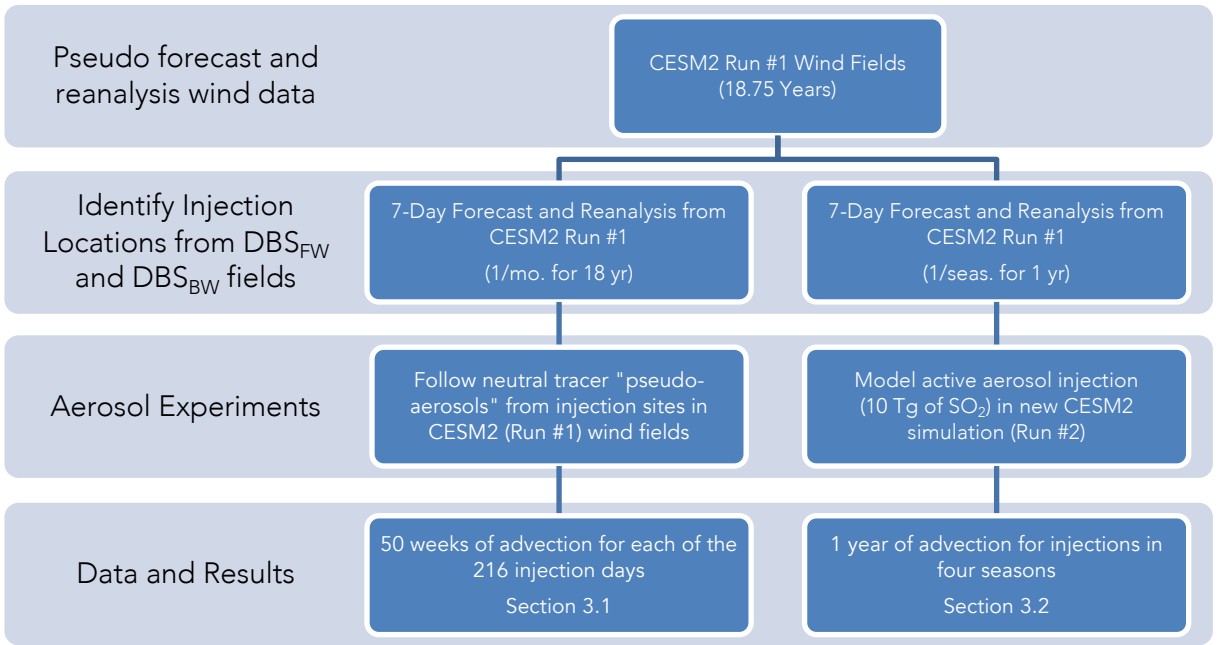

**Figure 3: Flow chart for geoengineering experiments**

A preliminary dispersion analysis was first conducted by approximating aerosol concentration evolution from the behavior of neutral tracers (pseudo-aerosols) that perfectly follow the wind fields (Figure 3, left column). At the beginning of each month for the full 18.75 years of CESM2 (WACCM6) model simulation, injection locations were identified using a short temporal neighborhood of the wind-field output from CESM2 (WACCM6) Run #1. The advection of parcels of neutral tracers from neighborhoods surrounding those injection points was then computed for the following 50 months in the Run #1 wind fields. This approximation of transport by perfectly fluid-following particles inherently assumes that there are negligible inertial effects and the vertical motion is not influenced by radiative heating or cooling of the particle (or gas). While these assumptions limit any study of climate impacts, these calculations provide a longitudinal comparison of dispersion from dynamics-informed injections and traditional injection protocols that spans multiple modes of interannual climate variability.

We complement our neutral tracer trajectory analysis with four comprehensive CESM2 (WACCM6) simulations spanning one-year after sulfate precursor injection (Figure 3, right column). Each simulation corresponds with injections during a particular season. These simulations incorporate the advection of aerosols with full microphysics, atmospheric chemistry and radiative forcing components, as well as all other earth system model components. Again, the performance of DBS-informed and fixed-location sites are compared. As the inclusion of microphysics and atmospheric chemistry makes these simulations computationally more expensive, no further improvements to injection site selection methods were evaluated, though several suggestions for future work are discussed in the discussion and conclusions.

We note that although Run #1 involves calculation of neutral tracers (resembling infinitesimal radiatively inert aerosols), Run #2 involves injection of the gaseous aerosol precursor $SO_2$. $SO_2$ requires time to convert to sulfate aerosols (e.g., Mills et al., 2017), and the injection strategy of $SO_2$ (for example along a longitudinal band instead of into a single grid box) has been demonstrated to affect aerosol size and hence radiative effects of the injection (e.g., English et al., 2012). Nevertheless, the purpose of these DBS-informed simulations is to describe the effects of recognizing transport barriers or atmospheric features that enhance transport. The applicability of this method is not dependent on whether a gas or particle is injected.

**2.2 Lagrangian transport extremizers**

Diffusion Barrier Strength (DBS) is an objective (i.e. observer-independent) diagnostic field whose ridges highlight diffusive or stochastic transport extremizers from velocity data [Haller et al., 2018]. For a given time-varying velocity field $\mathbf{v}(\mathbf{x}, t)$, and tracer $c(\mathbf{x}, t)$ we can describe the evolution of this tracer with the classic advection-diffusion equation:

$$\frac{\partial c}{\partial t} + \nabla \cdot (c\mathbf{v}) = \nu \nabla \cdot (\mathbf{D}\nabla c), \quad c(x, t_0) = c_0(x),$$

where $\mathbf{D}(\mathbf{x}, t)$ is the symmetric, positive definite diffusion-structure tensor. The left-hand-side of this differential equation contains the advection of this scalar field whereas the right-hand-side describes transport due to diffusive processes.

Furthermore, we define the path of a fluid particle in the velocity field $\mathbf{v}(\mathbf{x}, t)$ as a solution to the ordinary differential equation $\dot{\mathbf{x}} = \mathbf{v}(\mathbf{x}, t)$, described by the flow map:

$$\mathbf{F}_{t_0}^t(\mathbf{x_0}) = \mathbf{x}(t, t_0, \mathbf{x_0}).$$

From here, we define the DBS at a point $\boldsymbol{x_0}$ over the time interval $[t_0, t_1]$ as

$$DBS(\mathbf{x_0}) = \text{trace } \overline{\mathbf{T}}_{t_0}^{t_1}(\mathbf{x_0}),$$

where overbar denotes the time-average of the transport tensor

$$\mathbf{T}_{t_0}^t(\boldsymbol{x_0}) = \left[\nabla \mathbf{F}_{t_0}^t\right]^{-1} \mathbf{D}\left(\nabla \mathbf{F}_{t_0}^t, t\right)\left[\nabla \mathbf{F}_{t_0}^t\right]^{-\top},$$

for $t \in [t_0, t_1]$.

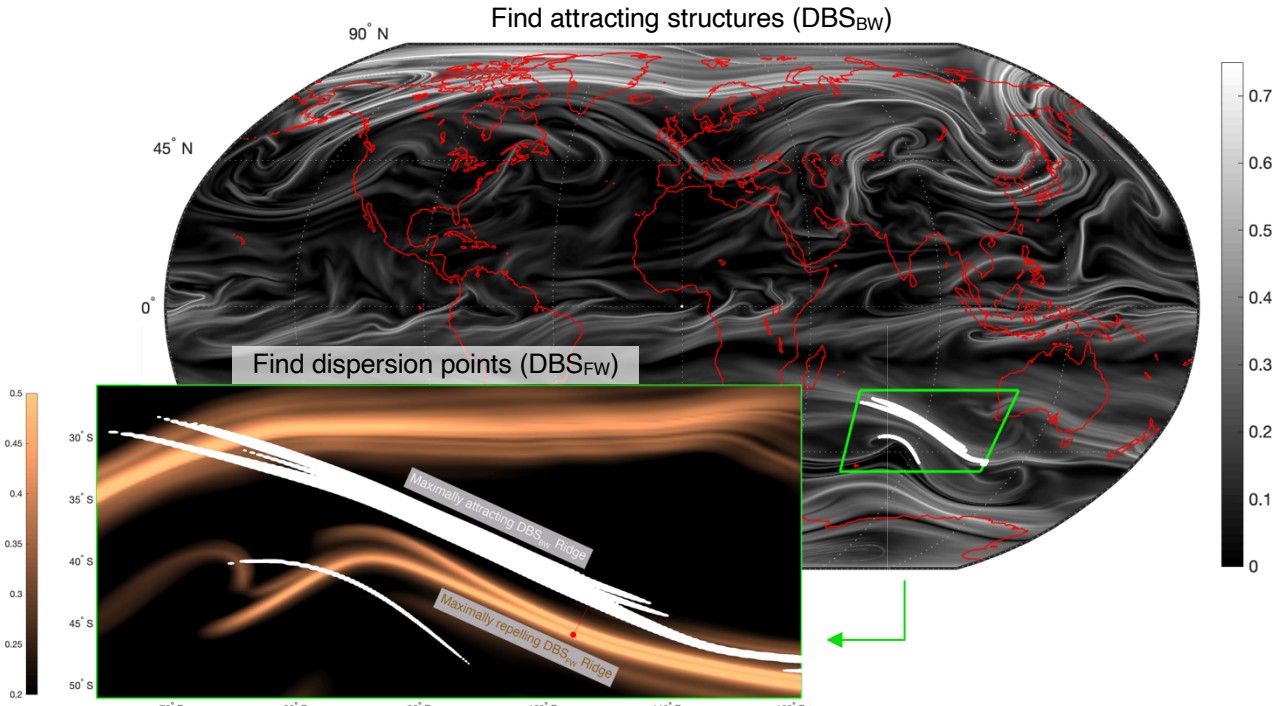

**Figure 4: Example of DBS-informed injection scheme at 540K that selects the injection sites. The global view shows seven-day DBSBW fields with two sections of disconnected strongly attracting structures highlighted in the green box. For the larger structure, we then identify all points closer to that attracting structure and select the unique point that will result in the most significant dispersion of aerosols. This is injection site is shown as the red dot on the DBSFW ridge in the inset. Injecting aerosols at these points will cause them to both spread quickly and converge to a large and complex attractor. Units for both forward and backward DBS fields are day⁻¹.**

The diffusion-structure tensor $\mathbf{D}$ is capable of representing parameterizations of many complex diffusion-like processes, but our research focuses on molecular (i.e. homogeneous, isotropic and steady) diffusion, in which case $\mathbf{D}(\mathbf{x}, t)$ is constantly the identity matrix. In this situation, the transport tensor $\mathbf{T}_{t_0}^t$ reduces to the inverse of the Cauchy-Green strain tensor, $\mathbf{C}_{t_0}^t =$

180 $\left[\nabla \mathbf{F}_{t_0}^{t}\right]^{\top} \nabla \mathbf{F}_{t_0}^{t}$, that also arises in the computation of the Finite-Time Lyapunov Exponent (FTLE) used in previous atmospheric transport barrier studies [see, e.g., Beron-Vera et al., 2012; Olascoaga et al., 2012; Garaboa-Paz et al., 2015; Serra et al., 2017; Wang et al., 2017]. DBS values are, therefore, pointwise equal to the trace of the time-averaged $[\mathbf{C}_{t_0}^{t}]^{-1} = \left[\nabla \mathbf{F}_{t_0}^{t}\right]^{-1}\left[\nabla \mathbf{F}_{t_0}^{t}\right]^{-\top}$ tensor. One notable difference between DBS and FTLE is the inclusion of diffusive or stochastic transport in the definition of transport barriers or enhancers for DBS, a process essential to predicting aerosol dispersion in the stratosphere. The inclusion

185 of diffusion in the transport functional allows for a systematic search for extremizing surfaces to transport [Haller et al., 2018], thereby eliminating the ambiguity inherent in various available coherent structure definitions [see Haller, 2015], or a lack of precisions from simple heuristics. Accounting for diffusive and stochastic transport necessarily leads to the inclusion of $\mathbf{C}_{t_0}^{t}$ tensors for all $t \in [t_0, t_1]$ in the definition of the DBS. In contrast, computing the FTLE only includes the single tensor $\mathbf{C}_{t_0}^{t_1}$.

Using a limited time-window of the modelled wind flow for DBS calculations, we were able to effectively simulate a real-time

190 geoengineering scenario. For each injection time, $t_0$, in our 18.75 years of simulation (Run #1), we analyzed one week of future flow data and one week of previous flow data as proxies for forecast and reanalysis, respectively, to determine optimal locations for sulfate injection. The one-week DBS$_{FW}$ field was calculated from $t_0$ to $t_0 + 7$, and under reversal of the direction of the flow in the reanalysis data, the DBS$_{BW}$ field was calculated from $t_0$ to $t_0 - 7$. As is described by Haller et al. [2018], the ridges of DBS$_{FW}$ highlight locations of strongest dispersion (i.e. diffusive transport limiters) on the globe at $t_0$, while the

195 ridges of DBS$_{BW}$ indicate the locations of the strongest accumulation (i.e. diffusive transport enhancers) at $t_0$. These diffusive transport barriers are analogous to the structures $M_A$ and $M_R$ from Figure 2, but account for diffusive as well as advective transport in the flow. To identify DBS ridges, we advected fluid particles along isentropic surfaces to simplify calculations and ignored vertical motions.

| **DBS-Enhanced Aerosol Injection Location Search Algorithm** |
|---|
| **Input**: Wind fields surrounding injection day ($t_0$) from $t_0$-7 to $t_0$+7 days. |
| 1) Calculate reanalysis DBS$_{BW}$ from $t_0$ to $t_0$-7, and forecast DBS$_{FW}$ $t_0$ to $t_0$+7.<br>2) Extract attracting ridges as connected components of DBS$_{BW}$ field above a fixed threshold via flood-fill algorithms.<br>3) Find seven largest ridges, and identify all points that are closer to each ridge than any other ridge.<br>    a. If we cannot find seven unique ridges, use as many unique ridges as we can, and separate ridges into intersections with latitude bands. Find points closest to our subdivided ridges.<br>4) If specified, restrict neighbourhood of ridges to that which intersects with neighbourhood of airports.<br>5) Identify the points with the highest DBS$_{FW}$ value for each neighbourhood and select the highest seven. |
| **Output**: Seven aerosol injection locations optimized for the wind flow on day $t_0$ |

200 **Table 1: Summary of the method of identifying injection locations for DBS-informed injections.**

We identified strongly attracting flow features as connected components of the DBS$_{BW}$ field with values above a simple fixed threshold. This threshold was chosen empirically from the range of DBS values in these calculations and was constant for all structure identification at all $t_0$. As also seen for other objective coherent structures identified from short-term calculations [e.g., Serra and Haller, 2016], these seven-day attracting features persist for much longer than their domain of computation in the flow and continue to attract many nearby fluid particles. Near each strongly attracting feature, the location with the largest DBS$_{FW}$ value signals a potential injection site for geoengineering as it indicates the strongest local dispersion over the next seven days. We balance strong dispersion and nearby strong attractors to both maximize the spread of aerosols and to prevent multiple injections being attracted to the same sections of the same attractor. When possible, this methodology prevented aerosols or precursors injected at initially distant sites from traveling great distances only to be attracted to the same portion of the flow. A flow chart detailing the injection-location selection process is shown in Figure 4.

While we prioritize injecting near unique attractors, this was not always possible given that single $DBS_{BW}$ ridges could also span much of the globe, and in rare instances strong attractors were not present in all regions. If seven unique attractors are not available at a given time, we simplify the process and choose the maximal $DBS_{FW}$ site near an attractor for each of seven latitude bands: [-7.5°, 7.5°], [±7.5°, ±22.5°], [±22.5°, ±37.5°], [± 37.5°, ±62.5°]. This dynamics-based injection approach, referred to as DI in the text, adapts to any isentrope or future climate scenario as the injection location choice always depends on the state of the stratosphere at the time of injection. This automated search algorithm is summarized in Table 1.

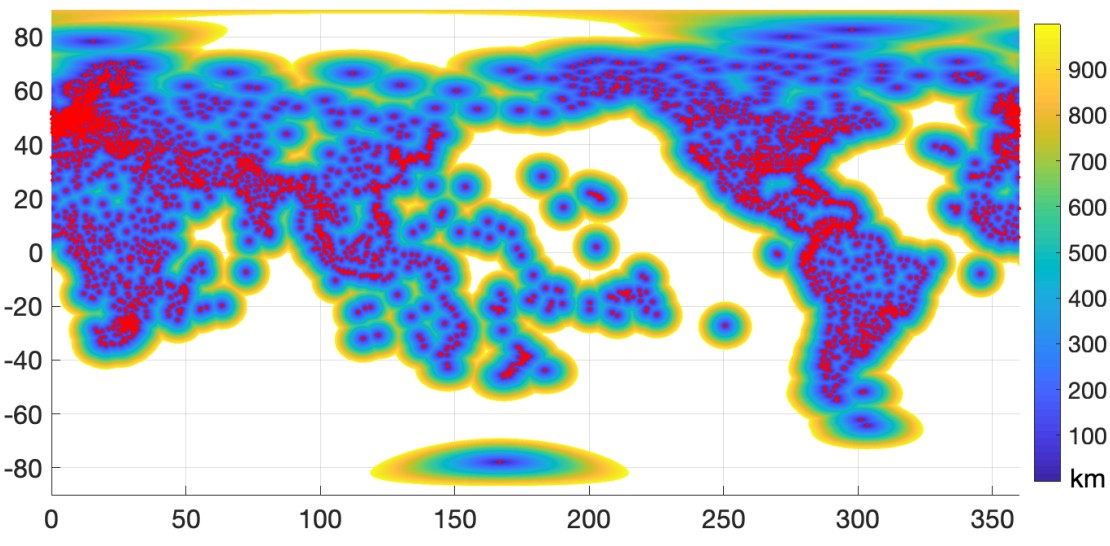

**Figure 5: Global coverage of potential injection locations for an airport-bound scenario including a map of 9300 airport locations (red dots) and the distance to the nearest airport up to 1000 km.**

As a control study, we ran a baseline scheme that injected sulfate aerosols at seven fixed-injection locations, referred to in the text as FI, $(0, \pm 15, \pm 30$ and $\pm 50°$ latitude at $260°$ longitude) similar to those explored by others [e.g. Robock et al., 2008; Heckendorn et al, 2009; Tilmes et al., 2017]. Lastly, we ran a scenario where DBS-injections were restricted to within 1000 km of an airport (scenario ADI in the text) [Global Airport Database, 2020] as a logistical handicap more similar to real world possibilities (Figure 5). For both the unrestricted DBS and the airport protocols, we limited the selection of injection locations to latitudes between $\pm 62.5°$ to avoid trapping by meridional barriers near the poles [Beron-Vera et al., 2012] while maximizing global coverage. Despite this restriction, the stratospheric flow sufficiently mixed aerosols across the globe, as with the FI experiments.

## 2.3 Geoengineering performance metrics

For our basic dispersion analysis, we evaluated the effective global coverage and rate of dispersion via an average minimum-distance metric defined as

$$\mu(t) = \frac{1}{N} \sum_{\boldsymbol{x} \in D} \min \left( d(\boldsymbol{x}, \gamma(t)) \right), \tag{1}$$

where $\boldsymbol{x} \in D$ are all points on the globe, $d$ is the great-circle distance, $\gamma(t)$ is the location of all neutral (pseudo-aerosol) tracers at time $t$, and $N$ is the number of grid points on the globe used for the calculation. Lower values of $\mu$ indicate a shorter distance from any point on the globe to the nearest pseudo-aerosol tracer, and thus imply better coverage. As volumetric or mass concentrations of aerosols are driving factors in many of the microphysical processes governing aerosol lifespan, we also calculated the entropy of the distribution of the pseudo-aerosols in our infinitesimal neutral tracer experiment. For a given probability $p_k$ on a discretized grid of (unequal) bins (such as tracer concentration),

$$E = -\sum_{\boldsymbol{x} \in D} p_k \log_2 \left( \frac{p_k}{w_k} \right), \tag{2}$$

where $w_k$ is the size of a bin [Harris, 2006]. The evolution of the entropy of each injection protocol was normalized by the entropy of a perfectly uniform distribution on the same discrete grid to give a normalized entropy value in the interval [0, 1]. At the beginning of each month during the 18.75-year CESM2 simulation, we initiated advection of fluid-following neutral tracers from seven DI sites, seven ADI sites, and seven FI sites that lasted for 50 weeks. In these initial experiments, we did not run a new simulation of CESM2(WACCM6), but used the advection of neutral tracers in wind fields generated by CESM2(WACCM6) (Run #1) to approximate the dispersion dynamics of aerosols in a fully turbulent stratosphere (left column of Figure 3).

In our second round of experiments, we used the precomputed wind fields from CESM2 Run #1 to determine injection sites, and then ran new CESM2 simulations starting in each season (Run #2) with 10 Tg SO₂ injections that included fully-coupled microphysics. In this way, the atmosphere was influenced by geoengineering in Run #2, but not in our neutral-tracer

experiments. The effective global coverage, SO₄ burden, and effective radii were then compared for the two DBS-informed (DI and ADI in the text) and one fixed (FI) injection protocols.

## 3 Results

### 3.1 DBS influence on pseudo-aerosol dispersion

For the infinitesimal neutral tracer advection experiment (Figure 3, left column), the global coverage of pseudo-aerosols injected at seven dynamically varying DBI locations was much greater than coverage from the seven fixed (FI) locations. We found an immediate increase in global coverage for the DI experiments, as predicted from the mathematical definition of large $DBS_{FW}$ values. Zonal concentrations of pseudo-aerosol tracers were calculated as the fraction of the total number of tracers present in a given discrete latitude band.


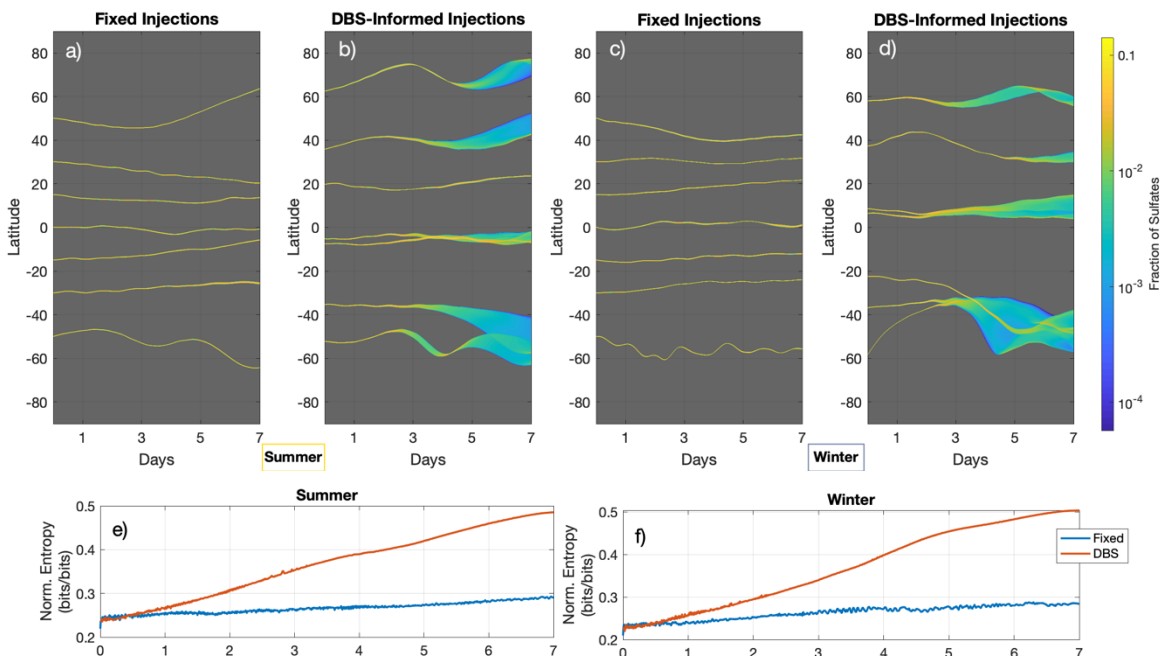

**Figure 6: DBS-informed injection yields significantly enhanced coverage over fixed-location injections over short-term, seven-day periods. Zonal concentrations in subplots a-d are calculated as the fraction of the total number of neutral-tracers (pseudo-aerosols) in a given latitude band at a given time. The time evolution of zonal concentration over one week of transport from the two injection protocols are displayed in subplots a-d with their respective normalized**
**entropy values in subplots e-f.**

Figures 6a and 6c detail how the zonal concentrations of these idealized sulfates injected at the standard FI sites evolve over the first seven days of transport during boreal summer and winter, respectively. While there is North-South meandering of the injected tracers, the fixed-injection scheme resulted in little-to-no dispersion by the end of the first week. In contrast, after only
three days of transport, Figure 6b and 6d show that the DI tracers have begun efficiently spreading and increasing global

coverage. As is discussed later and exhibited in the full microphysics simulations in the next section, this immediate dispersion (which while idealized in Run #1, could apply to aerosols or their gaseous precursors) has an impact on the rate of coagulation, sedimentation, and the effective radii and lifespan of the aerosols. By the end of seven days, the DI tracers have covered a large portion of the northern and southern hemispheres from $-70°$ to $70°$ for both the summer and winter injections. This

difference in global coverage between DI and FI schemes is further quantified by the normalized entropy of pseudo-aerosol tracer distributions for the two protocols. In the bottom two subplots of Figure 6 (6e and 6f), the pseudo-aerosol tracer distribution from the DI protocol has greater entropy (Eq. 2) after one day of transport with that performance gap widening for the entire week. At the same time, the near constant entropy for the FI experiments verifies that those clusters of neutral tracers have not yet dispersed, and create a longer window of time for sulfate "hot-spotting" and coagulation.


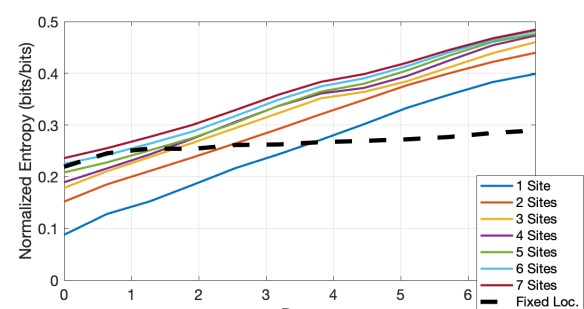

**Figure 7: Normalized entropy of DBS-informed injections under a varying number of sites for the summer simulation in Figure 6. Through optimizing injections near dispersion enhancing transport barriers, we are able to achieve significantly more uniform distributions of aerosols with fewer necessary injection sites.**

The enhanced dispersion, made possible by harnessing DBS information, also allows for a streamlining of injection operations. Using the same time period from Figure 6a-b, we were able to leverage the improved distributions of pseudo-aerosols and test how reducing the number of DBS injection sites would influence the subsequent global coverage. Figure 7 shows that for one such test, almost immediately, there is a negligible reduction in entropy when reducing from seven DBS-informed injections to six. That is, within the first day of dispersion, reducing the number of injection locations, and the amount of injected material

by nearly 15% does not impair the performance of our DBS protocol to levels below that of the fixed locations. After three days, when the influence of strong DBS barriers has been more effective, one can reduce injections to only two DI sites and still obtain a more uniform concentration distribution than with seven FI sites. From four days to the end of the first week, a single injection site was dispersing pseudo-aerosols in the stratosphere more effectively than the combination of all seven fixed sites. Not only could a well-informed choice of injection locations provide significant benefits for increasing concentration

homogeneity (thereby more evenly influence radiative forcing and reduce hot-spotting), there can be significant strategic and economic advantages of DBS-informed geoengineering programs.

To determine if injecting at DI sites would consistently increase dispersion over all seasons, and over many years, we consider the cumulative statistics of many long-term advection models. At monthly intervals, the same seven-day reanalysis and forecast method was used to choose DI locations, both with and without a 1000 km distance restriction to the nearest airport (ADI). After 1 week, 10 weeks, and 50 weeks of transport, $\mu$ values (Eq. 1) were computed and compared to the FI protocol. Figure 8 shows the results of this experiment for transport periods spanning the whole 18.75 years. Clusters of $\mu$ values indicate variance in the response of pseudo-aerosol transport to different DBS ridge structures over time, but mean values of those clusters (indicated by horizontal lines) consistently show improved coverage compared to the FI protocol. As noted before, the most considerable enhancement in dispersion was seen immediately, supporting the potential for this approach to influence aerosol microphysics during the first week of transport. After 10 weeks, DI injections were still more effective at global coverage than the FI protocol, even with the airport restrictions, but at yearly timescales, the average improvement was minimal. It should be noted that the variance of global coverage was also lowest for DI seeding at 10 and 50 weeks.

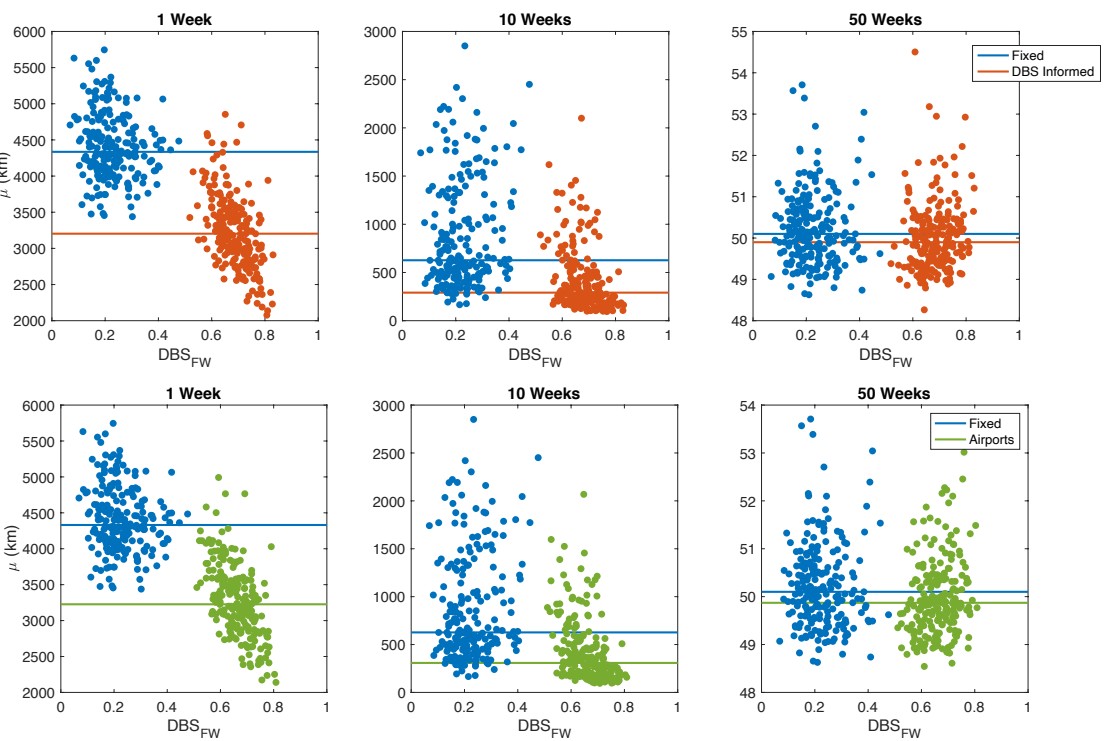

**Figure 8: Average distance to nearest aerosol (Eq. 1) with injections initialized each month for 18.75 years. The top three subplots compare the fixed location (FI) protocol to the DBS-informed (DI) injection after 1 week, 10 weeks and 50 weeks of transport, with cluster means marked by respective horizontal lines. The bottom three subplots are analogous with the added restriction that DBS informed injections must also be within 1000 km of an airport (ADI). Both DBS approaches outperform the fixed injections protocols up to 10 weeks, suggesting flexibility of the protocols and utility of harnessing Lagrangian coherent structures for enhancing dispersion.**

**3.2 Full atmospheric chemistry and microphysics simulations**

Beyond improved advective transport of aerosols or precursors, we also wish to investigate the role that diffusion transport barriers may play in dampening microphysical processes that can reduce the lifespan of geoengineering aerosols, such as coagulation and sedimentation, in a fully coupled climate model. To address this, we applied the DI and ADI site selection methods at the beginning of four months (January, April, July, September) during one year of CESM2(WACCM6) output. We then reran twelve CESM2(WACCM6) simulations with injections of 10 Tg of $SO_2$ for the three separate protocols, on a given day, $t_0$, in each season. For each model run, the $SO_2$ was divided evenly between the seven fixed or dynamic injection sites on the 540 K isentrope. This provided twelve year-long model simulations that calculated the total evolution of injections from each geoengineering protocol. The average effective radii of the resulting sulfate aerosols, total column $SO_4$ burden (kg/m$^2$), and top-of-atmosphere radiative forcing was measured on a lat-long grid over isentropes from 360 to 720 K. The seasonal experiment names referred to in this section correspond with the boreal season.

3.2.1 Aerosol Burden

As DBS ridges and this particular coherent structure view of stratospheric dynamics are mathematical tools to address dispersion and transport, we initially focus on enhancements in $SO_4$ dispersion and global coverage when using DBS-informed site selections. To account for the natural variability of $SO_4$ burden in our control runs, effective coverage was quantified from the cells whose total column $SO_4$ burden exceeds five times the average global burden for the one week prior to sulfate injection. The amount of global coverage is then the percent of the surface area of the earth with $SO_4$ exceeding this threshold. Around 1% of the surface area of the earth exceeds this threshold prior to injection.

Figure 9 shows the difference in global coverage between the DBS-schemes and the FI protocol. A consistent short-time pattern was evident in these time series for all four seasons' injection. There is an immediate positive difference with the DBS approaches as a greater percent of the Earth is efficiently covered by an above-average $SO_4$ burden. This initial improvement in coverage peaks between one week and two weeks after injection and is attributed to high $DBS_{FW}$ values at injection locations and an enhanced ability to strategically spread along nearby jets and eddies that were present in the $DBS_{BW}$ fields. These dispersion patterns and their correlation with $DBS_{BW}$ ridges can be seen in the $SO_4$ burden plots of Figure 10. This immediate improvement can be as high as 5% more global coverage, equating to a change in net radiation over an additional 32 million km$^2$, or more than the equivalent surface area of North America.

After the initial peak improvement in global coverage, there is often a rebound in Figure 9, at which point the FI aerosols can cover up to 12% more of the globe. Surprisingly, after this local minimum, there is always a secondary peak, sometimes larger than the first, showing a response in global coverage using the DBS methods well past the computational limitations of the

original DBS ridges. This second peak in performance occurs between six and ten weeks after injection, and enhanced coverage by the DBS methods can extend until all three experiments achieve total global coverage.

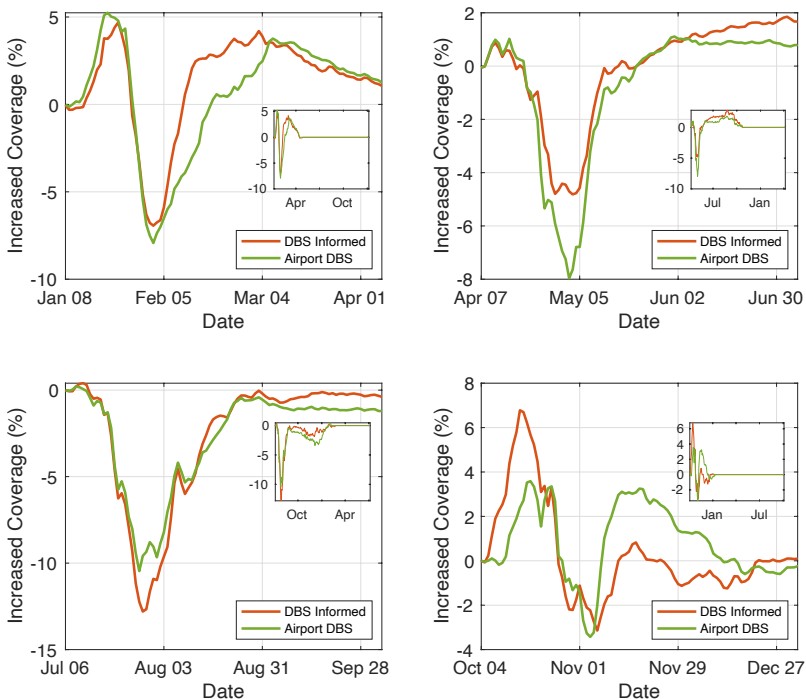

**Figure 9: Analysis of CESM2 (WACCM6) output showing the increase in percent of the Earth's surface surrounded by an $SO_4$ burden greater than five times the global mean from the week prior to injection as compared to fixed injection protocols. Large subplots show the first 90 days after injection and the smaller subplots show the first full**
**year.**

Notably, the spring and summer season injections had much smaller relative improvements in their initial peaks in Figure 9. A closer investigation of the dispersion patterns in Figure 10 begins to explain why. The left three columns of Figure 10 show the $SO_4$ burden with the two inset percentages in each plot detailing the proportion of the respective hemisphere's (North or
South) surface area covered by five times pre-injection burden means. The right column shows the $DBS_{BW}$ field calculated for the 540K isentrope wind fields from the injection time $t_0$ to $t_0 + 7$ days so that the location of attracting structures coincides with the concurrent dispersion patterns. The winter injections occurred in the presence of strong attracting features in most latitude bands, and the DBS-informed methods were able to exploit these, especially in the northern hemisphere. Dispersion along these attracting features continued to enhance coverage for DI and ADI injections well after the snapshot in
Figure 10. During spring, the DBS-informed injections exploited the similar attracting features in the northern hemisphere (13% vs. 12% coverage) but in the southern hemisphere, attracting ridges around -50° blocked aerosols from migrating further south in all three experiments.

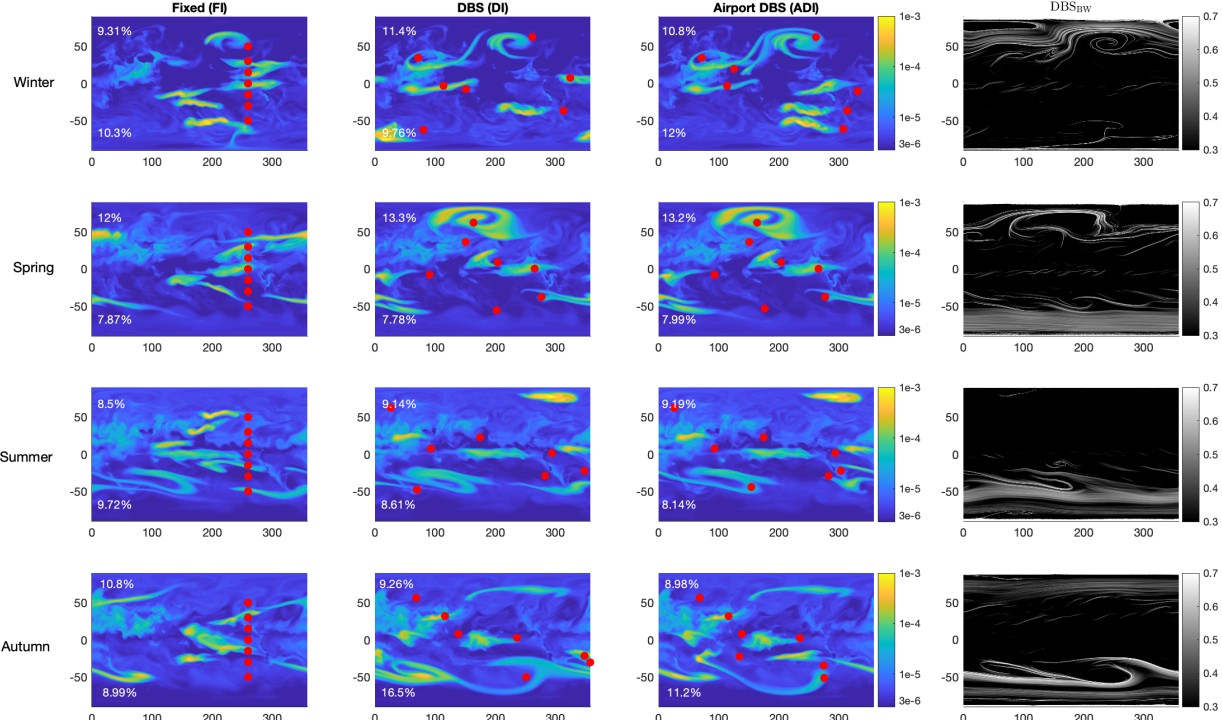

**Figure 10: Analysis of CESM2 (WACCM6) output showing the SO₄ burden after seven days of transport for the three injection protocols in each season. The percent of the earth's surface covered by an SO₄ burden greater than the global mean from the week prior to injection is noted in the top left of each panel. Original injection locations for each experiment are shown as red dots. Units for the colormap are kg m⁻².**


The summer injections occurred during an absence of strong attracting or repelling structures, except a dominant circumpolar feature in the southern hemisphere. The DBS_FW values for sites chosen north of -37.5° for the summer DI and ADI experiments were the lowest of all the experiments. In the northern hemisphere, aerosols spread by way of these locally maximal DBS_FW injection sites, but no strong anticyclonic structures, such as those found in the other seasons, were present. This prevented

northern hemisphere aerosol clouds from deforming along space-filling spiral features such as in the south and in other seasons. The autumn injection occurred during a time with stronger DBS_FW and DBS_BW ridges than the summer injection, and allowed for an enhanced dispersion in the south, especially for the DI experiment. The true strength of the DBS approach can be seen in the autumn experiment as only minor modifications in the southern hemisphere were necessary to achieve considerable enhancement in coverage. After seven days, DI SO₄ burden was above our threshold for 21% of the globe, versus only 16%

from FI. This advantage comes solely from enhanced performance in the southern hemisphere where DI coverage was 13.4%

and FI lagged at 6.8%. This significant advantage came from only a minimal change of injection point. The southernmost DI site was 0.25 degrees latitude further south than the FI site, and less than 650 km away, but the presence of strong $DBS_{BW}$ ridges, and complementary high $DBS_{FW}$ values allowed for a beneficial optimization.

Figure 11 details the $SO_4$ burden after eight weeks of transport. At this point, during the last oscillation of Figure 9 prior to total coverage, the three injection techniques begin to converge. Notable exceptions to this are the enhanced polar coverage in the winter DI injection in the autumn ADI experiment.

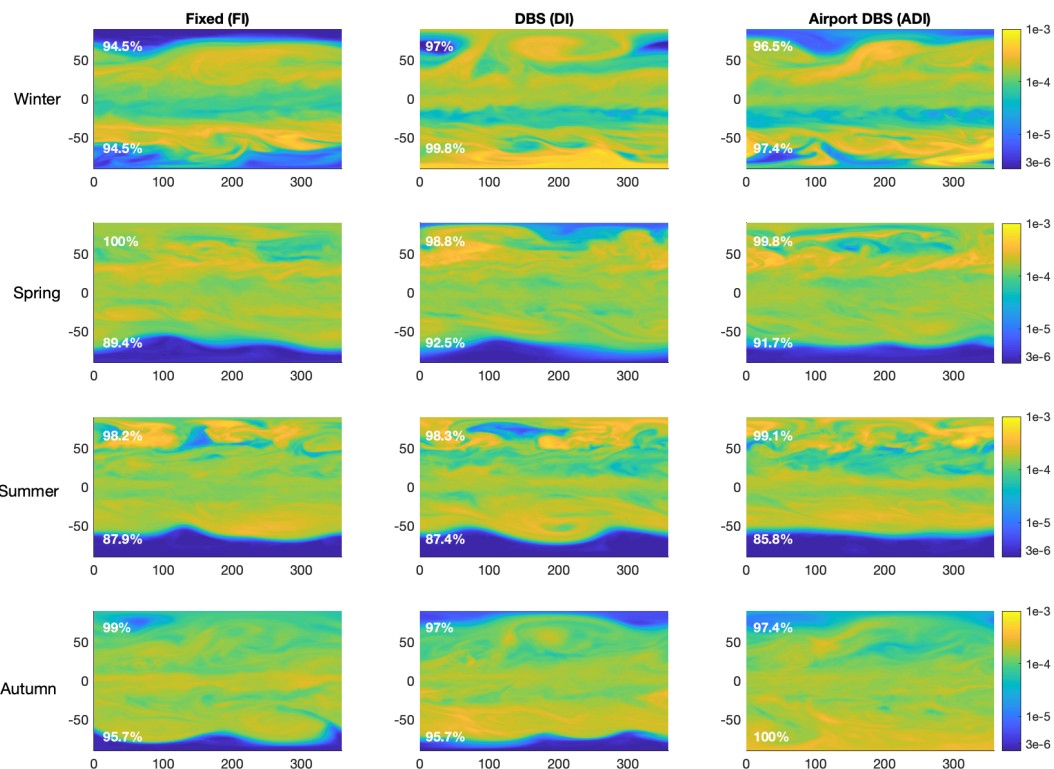

**Figure 11: Analysis of CESM2 (WACCM6) output showing the $SO_4$ burden after eight weeks of transport for the three injection protocols in each season. The percent of the earth's surface covered by an $SO_4$ burden greater than the global mean from the week prior to injection is noted in the top left of each panel. Units for the colormap are kg m$^{-2}$.**

### 3.2.2 Effects on Radiative Forcing

The dispersion patterns caused by the hyperbolic coherent structures in the stratosphere discussed in the previous section impacted the top-of-atmosphere radiative forcing (RF) in a complex way. The net shortwave and longwave fluxes were calculated for each grid cell on each day, as were the radiative fluxes for a control run over the same period without

geoengineering. The control fluxes were then subtracted from net fluxes to give a spatial and temporal distribution of the relative influence of each injection scheme. This change in RF is directly correlated with a change in temperature and is a strong indicator of the climatic influence of geoengineering [Hansen et al., 1997; Gregory et al., 2004].

Comparing the global effect of the FI protocol with DI and ADI, we find the cumulative impact of the DI and ADI injection was stronger in many cases. Table 2 shows the mean global change in RF for each injection protocol, during each experiment, after three periods of time (10 days, 30 days, 365 days), calculated as the average difference in net radiation at the respective time after injection. The three values in each column correspond to the global average FI (black), DI (orange), and ADI (green) difference from the control run in W m$^{-2}$. Gray shaded cells indicate times at which FI resulted in stronger radiative forcing

than DI and ADI. Bold DI and ADI values indicate a statistically significant difference (at 95%) in RF from the FI protocol using a two-sided $t$ test.

| | 10 Days | 30 Days | 365 Days |
|---|---|---|---|
| Winter | -1.4/-2.0/-1.7 | -6.9/-5.0/-6.2 | -7.6/-3.6/+0.4 |
| Spring | -1.1/-1.3/-2.0 | -7.6/-6.7/-6.1 | -11.1/-11.2/-9.6 |
| Summer | -1.5/-1.2/-0.8 | -4.0/-6.1/-6.6 | -3.4/-3.0/-2.0 |
| Autumn | -1.5/-2.0/-2.0 | -10.1/-10.4/-11.2 | -9.9/-11.1/-4.6 |

Table 2: Global average improvement in RF (W m$^{-2}$) at specified intervals after injection as compared to CESM2(WACCM6) control runs for FI, DI, and ADI injection schemes (left/middle/right resp.). Bold values indicate
a statistically significant difference in mean RF between FI and the corresponding DBS-informed injection on that day. Gray shaded cells indicate times at which FI resulted in stronger RF than both DI and ADI.

Over the first ten days of transport, the range of time for which our DBS methods can be mathematically supported, both global coverage and RF was often improved with DBS-informed injection. As could be expected from Section 3.2.1, there was a
reduction in RF for summer DI and ADI experiments. This corresponds with a lack of attracting and repelling structures, and questionable conditions for which to apply our injection site selection algorithm. After 30 days, only winter and spring RF for FI outperformed DI or ADI. This is during the rebound period detailed in Figure 9. At this point, well beyond the time horizon of our DBS calculations, summer and autumn DI and ADI had stronger RF than FI. After 365 of transport, FI outperformed the DBS protocols for the winter and summer injections. At these time scales it can be safely assumed that the chaotic nature
of stratospheric winds prevents any intelligible dependence on initial conditions for these injection experiments. There exists a complex nonlinear relationship between global coverage and RF, however, during the forecast windows we have investigated, there is a strong correlation between the enhanced dispersion from DBS-informed injections and RF. For longer term trends, one likely needs to be couple short time dispersion with other influential climatic variables, such as season of injection (e.g. Visioni et al., 2020).


3.2.3 Aerosol Effective Radii

The last metric from the geoengineered CESM2 simulations we analyzed is the effective radius of aerosols (Figure 12). The time evolution of the mass-averaged $SO_4$ aerosols was calculated on the 540 K isentrope, at the height where injection occurred. To prevent contributions of naturally occurring aerosols, the averages were calculated only using grid cells where the $SO_4$
burden exceed five times the pre-injection mean. During the winter season, the most dramatic change in radii occurred, with peak values for the simple injection protocol clearly exceeding the DI and ADI methods. Differences in other seasons were more minor, but the injection protocol peaked at higher values for both the spring and autumn experiments as well. During summer, there was reduced performance with the DBS-informed injections, as was also indicated in the RF and $SO_4$ burden analysis.

The improvement that was possible during the winter injection is notable as it suggests a better understanding of the connection between stratosphere dynamics and chemistry can clearly be beneficial for aerosol geoengineering. This is important because larger aerosols backscatter less (meaning more aerosol is required to achieve a given level of radiative forcing), heat the stratosphere more (resulting in greater side effects on stratospheric circulation and surface climate), and have increased sedimentation velocities (also meaning more aerosol is required) [Pierce et al., 2010; Tilmes et al., 2017; Simpson et al., 2019].

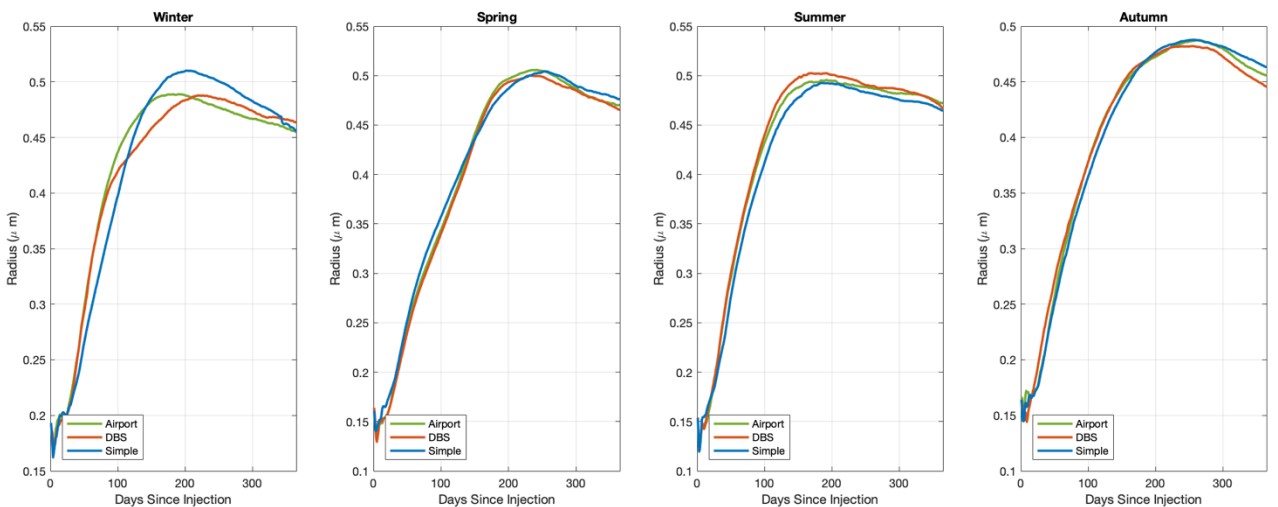

**Figure 12: Mass-averaged effective radius of injected $SO_4$ aerosols on the 540 K isentrope spanning one year of CESM2(WACCM6) simulations after injection.**

**4 Discussion and Conclusion**

Here we have explored the use of diffusive transport barriers to guide strategic injection locations for stratospheric aerosol geoengineering. Compared to commonly used methods that rely on fixed-injection locations, this dynamic site selection allows
for immediate improvements in particle dispersion and better global coverage, often with fewer injection sites. This has important implications for previous studies regarding the efficiency of aerosol optical depth versus injection rate. In particular, by focusing on only fixed-injection locations [e.g. Robock et al., 2008; Tilmes et al., 2017 among others] these studies

neglected an influential variable, the time-varying locations of stratospheric diffusive transport barriers. It is safe to assume that since these studies did not optimize injections for dispersion, they have thus far underestimated what is possible for geoengineered sulfate-aerosols reflectance. Using a Lagangian coherent structure informed approach via DBS fields shows promise for strengthening the role that geoengineered aerosols can play in altering climate dynamics, especially at short timescales or if logistical restrictions mean injection sites must be strategically chosen.

With dynamic injection locations, initial particle concentrations spread much more quickly as indicated in our tracer experiments (Figures 6 and 8) and in the full CESM2 simulations (Run #2, Figures 10 and 11). This reduces the probability of coagulation for each individual injection, and likely influenced lower peak effective radii present in DI and ADI experiments (Figure 12, e.g. Mills et al., 2017). With reduced coagulation, there will be slower descent of sulfates from the stratosphere (and out of action), as well as increased scattering. Second, increased dispersion uniformity, as quantified by normalized entropy in Figs 4 & 5, will reduce local stratospheric heating and result in more uniform radiative forcing. This is because heat transfer to aerosol particles in a given volume is proportional to the mass, which is reduced by lower concentrations. Lastly, in the fully-coupled CESM2(WACCM6) Run #2 experiments, DBS-informed injections improved global coverage by $SO_4$. This ability to more quickly achieve total coverage provides opportunity to strategize geoengineering protocols with a shorter window for interference in chaotic flows.

The results here indicate a predictable enhancement of dispersion for geoengineering if influential hyperbolic structures are present in the stratosphere. When there are strong short-term $DBS_{FW}$ and $DBS_{BW}$ ridges, such as in the winter, spring and autumn CESM2 Run #2 experiments, we show that we can exploit these ridges to optimize the immediate dispersion of aerosols. As well, the fine scale behavior of aerosol dispersion can be explained by the presence of influential structures, such as the attraction and blocking that occurs in the southern hemisphere Run #2 spring experiment. These fine scale structures have not been actively considered in geoengineering research, but may be exploited as is clear in the northern hemisphere for the winter and spring experiments (Figure 10). In our fully-coupled microphysics and atmospheric chemistry climate simulations, we also verified that initial improvements in particle dispersion from simplified flow calculations can result in less coagulation and increased aerosol spread as evidenced by more areal coverage by $SO_4$ burden and reduced effective aerosol radii. The enhanced global distribution of $SO_4$ for the two DBS-informed injection protocols after months of transport (e.g. Figure 11) speaks to the utility of strategic placement of aerosols or precursors near hyperbolic structures, as do long term radiative forcing improvements (Table 2), and correlated, yet complex, relationships with reducing average effective radii.

The extent of this influence is dramatically portrayed in the autumn CESM Run #2. At this time, a minimal modification of injection site in the southern hemisphere near strong hyperbolic structures, a change of less than 650 km, resulted in aerosols spreading over an additional 7.5% of that hemisphere after 7 days. After eight weeks, this immediate DI dispersion benefit was not as noticeable, but the ADI scheme still contributed to improved coverage over the FI aerosols. The enhanced coverage in the autumn experiment is furthermore coincident with a considerable improvement in RF for the DI experiments at 10, 30 and 365 days. Additionally, there was a minor but statistically significant reduction in average aerosol radii ($p < 1 \times 10^{-5}$) one year after $SO_2$ injection. At longer time scales for several of the other CESM2 Run #2 geoengineering experiments, the

relationship between immediate enhanced dispersion, radiative forcing, and aerosol radii was less clear. For example, in spring, the ability to achieve global coverage and reduce radiative forcing at long timescales was best for the DI protocol, but there was not a similar improvement in average aerosol effective radius. In the winter DI simulation, there was a significant reduction in aerosol radius and improved coverage compared to FI, but there was a weaker effect on radiative forcing one year after injection.

To manage uncertainties in atmospheric flow and climate response, several recent geoengineering climate modelling studies have employed a feedback algorithm that adjusts the $SO_2$ injection rate at one or more latitudes (Jarvis and Leedal, 2012; MacMartin et al., 2014; Kravitz et al., 2017) in response to changes in surface climate; studies to date have updated the forcing once per year. In future studies, this "slow" feedback could be integrated with DBS-informed injection. Outlining this process, every week of simulation, new injection locations would be determined based on wind fields from the previous week, using the DBS algorithm described previously to find locations within different latitude bands. The model would then be run forward for a week with the $SO_2$ injected at those locations. This process, which essentially constitutes a form of Model Predictive Control (Garcia et al., 1989), could be carried out for a year, at which point the injection rates to use in each latitude range would be updated using the same "slow" climate-response-dependent feedback.

This research has shown that adapting aerosol geoengineering injections methods by considering 2D Lagrangian coherent structures provides an obvious advantage for dispersion of aerosols by enhancing longer term dispersion dynamics from only short forecast data. All results in section 3.1 suggest DBS-informed sites reliably outperform fixed locations when considering aerosol dispersion along isentropes as is rigorously guaranteed in the DBS metric derivation. The site-selection algorithm developed herein, however, does not consider the full three-dimensionality of stratospheric flows, or any knowledge about common meteorological and climatic features. Thus, the user-independent injection protocol does not always result in enhanced radiative forcing or global coverage when strong attracting and repelling features are not present in all regions, such as the summer experiments in CESM2 Run #2. With this is mind, we suggest that future injection experiments use DBS fields and diffusive transport barriers to constrain their choice of injection site, but allow for user intervention in the absence of such strong dispersive ridges, and to consider other influential variables such as seasonality of injection (e.g. Visioni et al, 2020) and aerosol microphysics, such as temperature and humidity.

In one injection season (winter), there is an appreciable reduction in effective radius, and a more negligible effect in the others. This indicates that there is both potential for dynamic injection to result in smaller aerosols, and it suggests there is room for improving our understanding of the role dispersive stratospheric dynamics play in aerosol coagulation. Future work along these lines may further improve upon the findings indicated here as well as help to understand the limits of what improvement in reducing aerosol size is still possible by considering time-varying small-scale turbulent features.

Related to this study is the proposed idea of direct injection of $H_2SO_4$ droplets, instead of $SO_2$ gas, which would ostensibly create a more monodisperse particle distribution and thus delay coagulation [Pierce et al., 2010]. Further investigation is warranted to understand the relative effects of this method vs the $SO_2$ injection simulated in our CESM2 (WACCM6) simulations, particularly if injection locations are chosen dynamically. This is especially important given the stratospheric

chemistry involved in SO₂ injection, including the approximately one-month timescale of conversion from $SO_2$ to sulfate aerosols, although in principle a transport barrier would apply to both gases and particles.

The results presented here are for a single model; different models will indicate different stratospheric features and thus different transport barrier locations and strengths. Of key importance is that the long-term dispersion analysis and structure
identification methodology relied on two-dimensional transport along isentropes. This method has proven to be successful for advancing the goals of optimizing sulfate precursor injections; a full three-dimensional computation of DBS fields would further improve the results. Upon review of the results, it appears there was an over emphasis on the ability to separate "unique" attracting structures from the 2D isentrope data. With the automated algorithm defined in Table 1, pairs of injection sites were sometimes chosen to be close together as it appeared their injected aerosols would end up on separate structures. In fact, these
features may have actually been connected along the third dimension. Additionally, long-term trends of $SO_4$ burden present in the stratosphere were mixed, suggesting further considerations of seasonality (e.g. Visioni et al., 2020) and consideration of what structure is likely being represented by a DBS$_{BW}$ ridge (e.g. a jet stream, polar vortex, something much less substantial). Though not investigated in the present research, with the introduction of stratospheric heating, cross-isentropic flow is likely to occur (e.g., vertical uplift from the heating), potentially justifying a three-dimensional analysis for the flows used here.
Vertical transport of aerosol is likely inevitable, but a 3D DBS analysis would exponentially increase the complexity and the computational costs of finding injection locations. The currently proposed isentrope method is found to improve injection protocols at little to no increased operational cost as there are clear advantages in the short-time dynamics using the DBS forecasts. One alternative improvement to 3D DBS fields would be simultaneous 2D analysis of structures on a range of isentropes.
Several studies have found that the injection rate of SO₂ is the limiting factor in geoengineering efficiency by increasing coagulation [Heckendorn et al., 2009; Niemeier et al., 2011; Niemeier and Timmreck, 2015]. These studies, however, did not optimize the dispersion of SO₂ during the first days following injection, and therefore did not maximize the potential of sulfate injections and consequent radiative forcing in model simulations. We conclude that the exploitation of readily available short-range wind forecasts and reanalysis is a catalyst that will allow better understanding of what can be achieved with climate
geoengineering. It is possible that one of the reasons the improvements seen here are not more drastic is the acute focus on the response to large individual injections, a method not commonly used. We ran simulations that included a single day of injection in an effort to demonstrate dispersion capabilities. As the ability of DBS ridges to predict dispersion dynamics has now been shown, a logical next step is to pursue more climate focused studies, such as injecting less mass over many successive injections using concurrent predictions. While the use of DBS-informed injections does not address many of the potential hazards of
geoengineering [e.g. Robock et al., 2008; Heckendorn et al., 2009], it is an important step forward towards assessing the feasibility of geoengineering to prevent the climate from crossing a critical tipping point.

**Author Contributions**

NA, BK, DG and GH contributed to the writing of this manuscript and analysis of results. NA and GH developed the experiment design and methods. BK performed all CESM2(WACCM6) model simulations. NA performed all other tracer and DBS computations.

### Acknowledgments

Support for B.K. was provided in part by the National Science Foundation through agreement CBET-1931641, the Indiana University Environmental Resilience Institute, and the *Prepared for Environmental Change* Grand Challenge initiative. The Pacific Northwest National Laboratory is operated for the US Department of Energy by Battelle Memorial Institute under contract DE-AC05-76RL01830. This research was supported in part by Lilly Endowment, Inc., through its support for the Indiana University Pervasive Technology Institute, and in part by the Indiana METACyt Initiative. The Indiana METACyt

Initiative at IU was also supported in part by Lilly Endowment, Inc. The ocean flow data used for Figures 1 and 2 can be obtained from AVISO (http://www.aviso.oceanobs.com).

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
