# Peer review of "Harnessing Stratospheric Diffusion Barriers for Enhanced Climate Geoengineering"

_Atmospheric Chemistry and Physics, 2020_

## Referee Comment (RC1) · Anonymous Referee #1 · 12 Oct 2020

An interesting approach to identifying ideal aerosol-injection sites. The results convincingly show the the improvement obtained by finding optimal sites via the Diffusion Barrier informed approach compared to the fixed site approach. Furthermore, the authors not only look at just the wind fields, but re-do the microphysics calculations necessary to determine how the injected aerosols may affect the wind field.

The presentation needs some cleaning up and clarification:

- Line 28, missing citation

- Line 44, missing comma (turbulence coherence and mixing → turbulence, coherence and mixing)

- Line 63, says figure 4 but I think mean to reference figure 2.

- Line 145, the definition of DBS. It appears to be non-dimensional. Is that correct?

- Line 147, there are typos in the definition of $\mathbf{T}_{t_0}^{t}(\mathbf{x}_0)$, the superscript of $\mathbf{T}$ and the first argument of $\mathbf{D}$. The definition should be, $\mathsf{T}_{t_0}^{t}(\mathbf{x}_0) = \left[\nabla_0 \mathbf{F}_{t_0}^{t}\right]^{-1} \mathbf{D}(\mathbf{F}_{t_0}^{t}, t) \left[\nabla_0 \mathbf{F}_{t_0}^{t}\right]^{-T}$, Given the importance of this definition, I suggest not having it as an in-line equation; give it its own equation line.

- lines 151 and 153, typo in the superscript of $\mathbf{F}$

- Figure 3. This figure was a bit confusing. It took some time to realize that the background grayscale in the upper left figure was related to $DBS_{BW}$ whereas in the upper right it was $DBS_{FW}$. Also, why is it in a log scale, $\ln(DBS)/2t$, and what is $t$? The advection time-scale, so 7 days? What are the units of this log scale? Inverse days?

- Lines 176-185, This section seems to describe the identification of DBS-informed injection sites. But it does not appear to be automated. Does this involve a human-in-the-loop for each month during the two decade simulation time? Could this procedure be automated to optimize some cost function, such as the main two metrics given?

- Figure 5, the subplot labels (a),(b), etc are too small to see. The curves in the lower two plots are not labeled.

- Figure 7, DBS vs. Airport DBS: look very similar. How much did the airport restriction actually affect the choice of injection sites?

- line 308, it says Figure 8, but I think this is about Figure 9.

- Figure 10, does larger effective aerosol radius correlate to more coagulation? I am assuming this is the case but it was not stated.

- While it is good to see the mean distance quantity and entropy, how much of an effect are these things having on the actual reduction in global temperature? Sure the DBS informed location sites will spread out more and coagulate less and it is clear that will reduce the temperature but is not clear by how much. Would be helpful to know.

---

## Referee Comment (RC2) · Anonymous Referee #2 · 26 Oct 2020

The paper by Nikolas O. Aksamit and co-authors is investigating a method to identify dynamical injection locations for stratospheric aerosol geoengineering approaches. The paper demonstrates that within the first seven days, this approach produces a wider particle distribution than fixed injection locations, using a simplified model (without aerosol microphysics). In a second experiment, a fully interactive Earth System model is used to test this approach. The authors claim that this results in improved effects of this type of geoengineering application.

I have several problems with this paper, and I don't think that it can be published in the current from and a significant revision is needed.

1. From what is stated in the paper, the authors may wrongly assume that H2SO4 in the model refers to liquid sulfate (SO4=), which is an H2SO4 solution. However, in WACCM, H2SO4 is a gas that will nucleate and form sulfate aerosols. To derive the sulfate burden in WACCM, one needs consider use so4_a1, so4_a2 and so4_a3 variables. The lifetime of sulfate can be identified in using the excess sulfate burden to decrease by 1/e from its peak, where excess burden is the amount above the pre-injection burden. However, this can be tricky due to seasonal and natural variability.

2. I don't see any support for the conclusions that "studies appear to have under-estimated the potential coverage and therefore reflectance of geoengineered sulfate-aerosols". The coverage of sulfate aerosols has not been investigated in this paper based on the global interactive climate model. Also, the authors have not investigated changes in coagulation. They have been looking at the effective radius, which does not give a conclusive result. There are several other statements in the conclusions that have not been addressed in the paper, for example, this study has also not demonstrated that fixed aerosols result in "heating hot spots".

3. The dynamical injection method proposed here has demonstrated that within the first 10 days an idealized aerosol disperses faster if injected in regions in dispersion regions (which is not a new or surprising result). Any discussion about the efficiency of SO2 injections vs sulfate with regard to geoengineering is irrelevant, because the benefit has only been shown in a simple model and for a short time period. What happens if the new method is applied for 10 years and not just for 1 season? Will there be any difference in the aerosol burden?

4. Finally, it is not really clear how the approach of injecting into diffusive regions can improve controllability and improve a controller that relies on the fact that specific injection locations will result in specific AOD and temperature changes. Injecting into regions of increased mixing, how can this be used for the controller development?

Here are some more detailed comments (not all issues are listed)

Line 90: This part needs more explanation, since people should not be required to have to read the referenced papers.

Line 105: How were geoengineering objectives improved in this study? No temperature targets have been met with this method.

Line 111: What years has been chosen? The SSP8.5 experiment covers 2015-2100? Did you use a standard future fully coupled model configurations?

Line 118: The importance of transport barriers varies with altitude so differences with altitudes can be expected, what do you mean with "fundamental differences"?

Line 119: Please clarify how aerosols are modeled in this simulation, how do they become non-reactive fluid-parcels?

Line 121: Please explain what is meant by "short temporal neighborhood of the wind-fields output"? Also, the following sentence is unclear. Please describe, what you mean by fluid following particles (are those aerosols in the model, if so which aerosols where looked at?)

Line 126: Sentence is unclear: what "numerous natural climate cycles" is referred to here? Do you mean you used the entire 18.75 years for informing the injection locations?

Line 128: Are these 1-year simulations performed with prescribed wind fields? If so, are those wind fields derived from the 18.75 years of simulation? Having a table that describes the experiments would be helpful. What do you mean by seasonal injections of sulfate precursors? Did you inject SO2 only once or every season, how much?

Line 210-212: What do you mean by aerosol concentrations, what aerosols are used? What do you choose for the size of a bin, since this is a modal model?

Line 218ff: The text is very confusing. The authors discuss H2SO4 (the aerosol precursor gas) which then nucleates to form sulfate aerosols. After injections of SO2 and

oxidation to H2SO4, H2SO4 will decay and aerosol will be formed. It is not clear how the authors learn about the aerosol lifetime and microphysical processes, while investigating H2SO4.

Line 224: Please explain above how the model simulation has been performed without microphysics (do you mean aerosol microphysics)? Again, another very confusing statement here, the authors now state they injected aerosols. What aerosols have been used?

226: If this is a 2D model lon/lat, what are zonal concentrations? Zonal averages? Were injections performed at one point or over one longitude band? How fast are aerosols been transported longitudinal? How do those locations correspond to the transport barriers subtropics and polar jet stream? Why is there a difference between winter and summer?

228: What do you mean by: there is north movement of "the volume of particles"?

Figure 5: From the figure caption one cannot understand what the lines represent? Fraction of sulfates with regard to what?

Figure 6: Assuming that the same amount of "aerosol" has been injected for the fixed and the variable injection sides, how can the Relative Entropy be different at the start?

---

## Author Comment (AC1) · 5 Feb 2021

Thank you for your comments and suggestions. They have greatly added to the clarity of the manuscript.

**Line 28, missing citation**

Thank you. We have added the missing citation.

**Line 44, missing comma (turbulence coherence and mixing → turbulence, coherence and mixing)**

[Figure]

Corrected, thank you.

**Line 63, says figure 4 but I think mean to reference figure 2.**

Corrected, thank you.

**Line 145, the definition of DBS. It appears to be non-dimensional. Is that correct?**

Yes, this is non-dimensional.

**Line 147, there are typos in the definition of $T_{t0}(x0)$, the superscript of T and the first argument of D. The definition should be, ... , Given the importance of this definition, I suggest not having it as an in-line equation; give it its own equation line. Lines 151 and 153, typo in the superscript of F**

Thank you for catching these superscript typos. They are now corrected.

**Figure 3. This figure was a bit confusing. It took some time to realize that the background grayscale in the upper left figure was related to DBSBW whereas in the upper right it was DBSFW. Also, why is it in a log scale, ln(DBS)/2t, and what is t? The advection time-scale, so 7 days? What are the units of this log scale? Inverse days?**

Correct, the timescale is inverse days. The logscale provides a better visualization of the structures present in the DBS field, instead of only showing a few ridges. This presentation is adapted from the standard approaches of presenting non-diffusive analog, FTLE fields. The figure has also been updated for clarity.

**Lines 176-185, This section seems to describe the identification of DBS-informed injection sites. But it does not appear to be automated. Does this involve a human-in-the-loop for each month during the two decade simulation time? Could this procedure be automated to optimize some cost function, such as the main two metrics given?**

This process is in fact automated. The algorithm is as follows 1: Calculate $DBS_{BW}$ and $DBS_{FW}$. 2: Extract attracting ridges as connected components of $DBS_{BW}$ field above a fixed threshold via flood-fill algorithms. 3: Find seven largest ridges, and identify all points that are closer to each ridge than any other ridge. 3a:If we cannot find seven unique ridges, use as many unique ridges as we can, and separate ridges into intersections with latitude bands. Find points closest to our 7 subdivided ridges. 4: In the neighborhood of each ridge, identify the point with the highest $DBS_{FW}$ value.

This could indeed be optimized, and would possibly improve the results, especially for the case when we cannot find 7 unique connected components, or by using adjacent isentropes. At this stage, a true optimization would be a significant undertaking as it would require many more CESM2 simulations. This is already the bottleneck, and why we have chosen this DBS method to begin with.

This algorithm is now included in Table 1.

**Figure 5, the subplot labels (a),(b), etc are too small to see. The curves in the lower two plots are not labeled.**

Thank you. This figure has been updated for clarity.

**Figure 7, DBS vs. Airport DBS: look very similar. How much did the airport restriction actually affect the choice of injection sites?**

12 of the 28 injection sites were changed upon applying the airport restriction. During this process, a given injection location was changed anywhere between 600 and 18,000 km. Considering we are moving injection locations halfway around the world, it speaks to the ability of DBS to highlight locations of strong dispersion as the results with the airport restriction can still outperform the fixed locations.

**line 308, it says Figure 8, but I think this is about Figure 9. C2**

Corrected, thank you.

**Figure 10, does larger effective aerosol radius correlate to more coagulation? I am assuming this is the case but it was not stated.**

Yes this is now explicitly stated in Line 440.

**While it is good to see the mean distance quantity and entropy, how much of an effect are these things having on the actual reduction in global temperature? Sure the DBS informed location sites will spread out more and coagulate less and it is clear that will reduce the temperature but is not clear by how much. Would be helpful to know.**

We agree that some aggregate measure of the net effect would be useful. The simulations are not long enough to capture steady state responses of temperature. Instead, we include information on the top-of-atmosphere radiative flux changes in a new Table 2, which, in this setup, are good estimates of radiative forcing (Hansen et al., 1997). This can serve as a proxy for temperature change (Gregory et al., 2004).

References:

Hansen, J., M. Sato, and R. Ruedy, 1997: Radiative forcing and climate response. J. Geophys. Res., 102, 6831-6864, doi:10.1029/96JD03436. Gregory, J. M., Ingram, W. J., Palmer, M. A., Jones, G. S., Stott, P. A., Thorpe, R. B., . . . Williams, K. D. (2004). A new method for diagnosing radiative forcing and climate sensitivity. Geophysical Research Letters, 31(3), 2–5. https://doi.org/10.1029/2003GL018747

Please also note the supplement to this comment:
https://acp.copernicus.org/preprints/acp-2020-696/acp-2020-696-AC1-supplement.pdf

---

## Author Comment (AC2) · 5 Feb 2021

Thank you for your suggestions. They have greatly improved the manuscript.

**1. From what is stated in the paper, the authors may wrongly assume that H2SO4 in the model refers to liquid sulfate (SO4=), which is an H2SO4 solution. However, in WACCM, H2SO4 is a gas that will nucleate and form sulfate aerosols. To derive the sulfate burden in WACCM, one needs consider use $so4_a1$, $so4_a2$ and $so4_a3$ variables. The lifetime of sulfate can be identified in using the excess sul-**

[Figure]

**fate burden to decrease by $1/e$ from its peak, where excess burden is the amount above the pre- injection burden. However, this can be tricky due to seasonal and natural variability.**

Thank you for bringing this to our attention. We have replaced our analysis of H2SO4 with SO4 burden.

**2. I don't see any support for the conclusions that "studies appear to have under- estimated the potential coverage and therefore reflectance of geoengineered sulfate- aerosols". The coverage of sulfate aerosols has not been investigated in this paper based on the global interactive climate model.**

We have expanded on the analysis of aerosol burden in the global interactive climate model. The results in section 3.2.1 of this manuscript are concerned with the ability of the DBS-informed injection schemes to achieve better global coverage in the first three months of transport, which support our statement that other geoengineering model experiments have not optimized injections for efficiently achieving global coverage and maximizing reflectance.

**Also, the authors have not investigated changes in coagulation. They have been looking at the effective radius, which does not give a conclusive result. There are several other statements in the conclusions that have not been addressed in the paper, for example, this study has also not demonstrated that fixed aerosols result in "heating hot spots".**

We have now included comments that explicitly state that rates of coagulation are correlated with the effective radius, with more coagulation causing larger radii (L440) (e.g. Mills et al., 2017). We have clarified our statement about hot spotting (L265) and instead refer to homogeneous cover and more uniform radiative forcing, which we did address through normalized entropy analysis and global coverage.

**3. The dynamical injection method proposed here has demonstrated that within**

**the first 10 days an idealized aerosol disperses faster if injected in regions in dispersion regions (which is not a new or surprising result).**

This mathematically rigorous identification of the time-varying locations of maximal dispersion in the stratosphere is indeed new. As well, nobody has addressed the combination of enhancing dispersion and the role that global attractors play (backward DBS ridges) in constraining the spread of aerosols. While these injection locations may sometimes coincide with a user's intuition, there has been no mathematically rigorous effort to identify where these positions are at a given point in time, as all previous efforts rely on average long-time behavior or proxies for dispersion. The present research provides a way in which the geoengineering community can be more strategic and precise with their injections.

Perhaps the reviewer could provide us with a citation.

**Any discussion about the efficiency of SO2 injections vs sulfate with regard to geoengineering is irrelevant, because the benefit has only been shown in a simple model and for a short time period. What happens if the new method is applied for 10 years and not just for 1 season? Will there be any difference in the aerosol burden?**

Thank you for asking this. We have reframed our results to focus on aerosol burden.

As is stated in the manuscript, these experiments present the role that short-time diffusive transport barriers can play in dispersion of aerosols at much longer time-scales. Because this is an initial investigation, we did not want to complicate the results with multiple injections into the same model simulation and have compounding influences, as would likely be done in a real geoengineering application. As well, our method was not applied for one season. We are looking at the evolution from a single day of injections over the course of a year. After the span of one year, turbulence in the stratosphere has effectively mixed sulfates by all approaches, though there are still improvements possible in global energy fluxes as indicated in the new Table 2 and Figure

9.

We have also mentioned this explicitly in the results of the neutral tracer experiment in the following line (L292): *"After 10 weeks, DBS-informed injections were still more effective at global coverage than the fixed-location protocol, even with the airport restrictions, but at yearly timescales, the average improvement was minimal."*

With the inclusion of aerosol burden values, it is now mentioned in section 3.2 that there is significant difference in performance of the DBS-enhanced and fixed injection protocols. The current simulations are not available for a 10-year analysis.

**4. Finally, it is not really clear how the approach of injecting into diffusive regions can improve controllability and improve a controller that relies on the fact that specific injection locations will result in specific AOD and temperature changes. Injecting into regions of increased mixing, how can this be used for the controller development?**

There may be some confusion regarding this comment. We do not reference a controller in the manuscript and are uncertain what is meant by controller development. The present research is focused on increasing dispersion at short timescales in order to overcome previously described limitations in geoengineering associated with rates of injection and subsequent coagulation, as well as other downstream effects. We do mention now (L446) that having a shorter time window from injection to achieving global coverage provides less time for interference, such as the influence of unforeseen attracting structures or coagulation.

**Line 90: This part needs more explanation, since people should not be required to have to read the referenced papers.**

This section has been expanded as follows: Recently, Haller et al. [2018, 2020] derived an additional objective criterion that specifically identifies the strongest barriers and enhancers of diffusive particle transport. That is, one can identify the time-varying

locations of material barriers in a fluid flow that maximize or minimize the diffusive contribution in the advection-diffusion equations over a given timeframe. They have obtained a diffusion barrier strength (DBS) field whose ridges highlight the strongest diffusive transport barriers in forward-time fluid flow analysis and strongest diffusive transport enhancers by running a backward-time fluid flow analysis. Neither of these simulations require modeling the evolution of a diffusive scalar field, but still rigorously define the structures that are most influential to diffusive transport.

Additionally, more detail is included explicitly in the methods section 2.2

**Line 105: How were geoengineering objectives improved in this study? No temperature targets have been met with this method.**

We have rephrased the sentence as follows: *"We find significant improvement in the ability of injected aerosols to both quickly surround the earth, and to be able to achieve similar coverage with fewer injection sites."*

**Line 111: What years has been chosen? The SSP8.5 experiment covers 2015-2100? Did you use a standard future fully coupled model configurations?**

The analysis was performed on a simulation using 2015-2016 conditions.

**Line 118: The importance of transport barriers varies with altitude so differences with altitudes can be expected, what do you mean with "fundamental differences"?**

This part of the manuscript has been rewritten in the new 128-136:

*A preliminary dispersion analysis was first conducted by approximating aerosol concentration evolution from the behavior of neutral tracers that perfectly follow the wind fields (Figure 3, left column). At the beginning of each month for the full 18.75 years of CESM2 (WACCM6) model simulation, injection locations were identified using a short temporal neighborhood of the wind-field output from CESM2 (WACCM6) Run 1. The advection of parcels of neutral tracers from neighborhoods surrounding those injection*

*points was then computed for the following 50 months in the Run 1 wind fields. This approximation of aerosol transport by perfectly fluid-following particles inherently assumes that there are negligible inertial effects and the aerosol vertical motion is not influenced by radiative heating or cooling of the particle. While these assumptions limit any study of climate impacts, these calculations provide a longitudinal comparison of dispersion from dynamics-informed injections and traditional injection protocols that spans multiple modes of interannual climate variability.*

**Line 119: Please clarify how aerosols are modeled in this simulation, how do they become non-reactive fluid-parcels?**

Thank you for bringing this confusion to our attention. We have clarified this section in the new L128-136. See previous comment.

**Line 121: Please explain what is meant by "short temporal neighborhood of the wind- fields output"? Also, the following sentence is unclear. Please describe, what you mean by fluid following particles (are those aerosols in the model, if so which aerosols where looked at?)**

We have rephrased here and elsewhere so that it is clearer that "fluid following particles" are actually neutral tracers that do not influence the wind, and perfectly follow air particles in the stratosphere. See new L128-136 and previous two comments.

**Line 126: Sentence is unclear: what "numerous natural climate cycles" is referred to here? Do you mean you used the entire 18.75 years for informing the injection locations?**

We have rephrased this sentence as follows:

*"While these assumptions limit any study of climate impacts, these calculations provide a longitudinal comparison of dispersion from dynamics-informed injections and traditional injection protocols that spans multiple modes of interannual climate variability."*

**Line 128: Are these 1-year simulations performed with prescribed wind fields? If so, are those wind fields derived from the 18.75 years of simulation? Having a table that describes the experiments would be helpful. What do you mean by seasonal injections of sulfate precursors? Did you inject SO2 only once or every season, how much?**

Thank you for the suggestion. We have included a flow chart (new Figure 3) to detail the two experiments we ran, and how they are related to the 18.75 years of CESM2 wind field.

**Line 210-212: What do you mean by aerosol concentrations, what aerosols are used? What do you choose for the size of a bin, since this is a modal model?**

We have clarified that the definition of Eq 2 is with respect to infinitesimal neutral-tracer particles in the stratosphere. This is not a modal model, rather we are quantifying the rate of mixing in the stratosphere wind fields.

**Line 218ff: The text is very confusing. The authors discuss H2SO4 (the aerosol pre- cursor gas) which then nucleates to form sulfate aerosols. After injections of SO2 and oxidation to H2SO4, H2SO4 will decay and aerosol will be formed. It is not clear how the authors learn about the aerosol lifetime and microphysical processes, while investigating H2SO4.**

Thank you for the suggestions. We have changed our analysis to focus on SO4 burden.

**Line 224: Please explain above how the model simulation has been performed without microphysics (do you mean aerosol microphysics)? Again, another very confusing statement here, the authors now state they injected aerosols. What aerosols have been used?**

This point has been clarified by the inclusion of a flow chart (Figure 3) that defines the two separate experiments more clearly.

**226: If this is a 2D model lon/lat, what are zonal concentrations? Zonal aver-**

**ages?**

Zonal concentration refers to the fraction of the total number of tracers in each latitude band. This has been clarified in the text by adding the following sentence: Zonal concentrations of tracers were calculated as the fraction of the total number of tracers present in a given discrete latitude band. (L246)

**Were injections performed at one point or over one longitude band?**

Injections were performed at seven separate locations. This has been clarified in the following sentence (L243): *For the infinitesimal neutral-tracer advection experiment (Figure 3, left column), the global coverage of pseudo-aerosols injected at seven dynamically varying DBI locations was much greater than coverage from the seven fixed (FI) locations.*

This is also clarified in the description of our search algorithm in Table 1.

**How fast are aerosols been transported longitudinal?**

These "pseudo-aerosols" perfectly follow the wind fields along the 540 K isentrope.

**How do those locations correspond to the transport barriers subtropics and polar jet stream?**

Thank you for bringing this to our attention. We have now included DBS surface plots and SO4 burden fields at two weeks and six weeks after injection (Fig 11, 12). In these images, multiple transport barriers are present as is their influence on collecting or blocking aerosols. We also future work that could connect well known meteorological phenomena with DBS ridges (L503).

**Why is there a difference between winter and summer?**

These are time-varying wind fields carrying the neutral tracers that we are tracking. The difference in winter and summer can be easily attributed to the time-varying nature of this fluid flow.

**228: What do you mean by: there is north movement of "the volume of particles"?**

While there is North-South meandering of the injected tracers, the fixed-injection scheme resulted in little-to-no dispersion by the end of the first week.

**Figure 5: From the figure caption one cannot understand what the lines represent? Fraction of sulfates with regard to what?**

This has been clarified both in the text and in the caption:

*Figure 6: DBS-informed injection yields significantly enhanced coverage over fixed-location injections over short-term, seven-day periods. Zonal concentrations in subplots a-d are calculated as the fraction of the total number of neutral-tracers in a given latitude band at a given time. The time evolution of zonal concentration over one week of transport from the two injection protocols are displayed in subplots a-d with their respective normalized entropy values in subplots e-f.*

**Figure 6: Assuming that the same amount of "aerosol" has been injected for the fixed and the variable injection sides, how can the Relative Entropy be different at the start?**

Concentrations were calculated by the number of particles in a lat-long grid, and the area of each cell varies with latitude. As the "area" of the parcel of particles is smaller than the grid cells, having the same number of particles in a grid closer to the equator results in a different computation of concentration. The minor difference present between the "7 sites" and "Fixed Loc." in this graph is the result of this grid.

Please also note the supplement to this comment:
https://acp.copernicus.org/preprints/acp-2020-696/acp-2020-696-AC2-supplement.pdf
* * *

---

## Referee Report (RR1)

Review:
The authors have appropriately addressed most of my comments, but there are still questions and concerns about the paper that require substantial revision of the text.

First point: It seems like there is still confusion about injecting aerosols vs aerosol precursors and there needs more clarification and discussion throughout the text. The text often refers to aerosol injections, which is however not applied in most model studies and potentially for applications. Injecting aerosol precursors (not aerosols) will result in a different evolution and growth of aerosols, than aerosols. This is because the time it takes for nucleation of sulfur (SO2), as for example discussed in Mills et al. (2017). The lifetime of SO2 is usually about 30 days, but it depends on the availability of OH. After a large volcanic eruption, or large SO2 injections, OH depletion can lead to a prolonged lifetime of SO2 (47 days after MT Pinatubo). The application of the dispersion model using a tracer or aerosol is therefore different than injecting SO2. To outline the complexity, it has been shown that SO2 injections at a point location results in smaller aerosols than injections at a longitudinal band. This is because, the zonal wind is already dispersing the gas quickly and therefore reduce the amount at the injection location. More nucleation is induced instead of condensation on existing participles. Furthermore, after enhanced aerosol burden has been established in the stratosphere, sulfur injections will condensate on existing particles independent on the dispersion efforts.

Second point: From what is shown in the paper, I am not convinced that this method leads to significant improvements. As shown in the paper, after a one-year simulation of the fully coupled model, it seems that there are no significant differences in coverage between the fixed injection method and the dynamically derived injection method in terms of efficiency (Fig 9). Fig 9 is showing a strong reduction in coverage in the first 30 days, which may have to do with the lifetime of SO2, and how long it takes to build up a larger sulfate coverage. It would be helpful to also show the absolute values of sulfate here, which are likely to be very small initially. Furthermore, Table 2 indicates, that after 30 and 365 days, in 2 out of 4 cases, the radiative forcing is more strongly reduced in the fixed injection case and the third case shows almost no difference between the DI method. I agree that this method may be useful to consider for the onset of sulfur injections and should be explored in more detail. However, I don't think the authors can support that there are "long-term gains in sulfate burden and radiation" as stated in the conclusions. I would therefore at this point advise against recommending this method as "a benchmark improvement in injection protocol".

Finally, as stated in my earlier review, two co-authors of this study work on a feedback controller to improve climate impacts. It is important to address the question whether this method is suitable for applying a feedback-controller. Injection locations and amount have been chosen to improve the climate outcomes in particular of surface temperature. How could this method be integrated?

---

## Referee Report (RR2)

Review:

The authors have followed my suggestions regarding the first and third comment, but there are still some concerns regarding the second point. The authors state in their response that "the intended effect is harder to predict" in using the full Earth System model. As outlined in the earlier review, Table 2 does not show the expected improvement between the FI and the other cases, and at least half of the times the RF for the FI (fixed injection) case is larger than the other cases. As can be seen in the table for 365days, for winter and summer, the FI is showing a larger forcing (significantly larger in winter), while for spring and summer the RF is slightly smaller compared to the FI case (see Table 2 below). This should be mentioned in the text. One reason for this signal may be more complicated aerosol microphysics in the atmosphere, using SO2 injections than what can be estimated from a simple model. The method may be useful but has to be further investigated before recommending as an application for aerosol geoengineering.

The authors have added some minor changes to the wording in the Discussion and Conclusions. However, the abstract is still presenting results that are not reflected in the paper and therefore need to be changed: "*Additionally, this enhanced dispersion slows aerosol microphysical growth, reducing the effective radii of aerosols **at monthly timescales. This has long term impacts on radiative forcing, beyond the lifespan of the original influential transport barriers. We conclude that previous feasibility studies of geoengineering likely underestimate the potential cooling efficiency of sulfate aerosol geoengineering by not strategically injecting at optimized dispersion locations**.*"

1.  The authors have not shown that the changes (increase and reduction, and not only reduction) in the effective radii of aerosols (at one theta level that is mostly above the enhanced aerosol layer in high latitudes) would change the radiative forcing. For example, it seems like the effective radius at 540K in spring (in Fig12) is largest for the simple case, but the radiative change (Table 2) does not reflect that and has a reduced radiative effect compared to the other cases.
2.  The conclusion that the potential cooling efficiency of sulfate aerosol geoengineering is likely underestimated has not been shown in the paper.

The abstract would be much improved if the authors would point to the potential for this method, but also mention the need for further investigation of these methods in order to gain a more robust understanding.

|  | 10 Days | 30 Days | 365 Days |
|---|---|---|---|
| Winter | -1.4/*-2.0*/*-1.7* | -6.9/**-5.0**/*-6.2* | -7.6/**-3.6**/**+0.4** |
| Spring | -1.1/*-1.3*/**-2.0** | -7.6/*-6.7*/**-6.1** | -11.1/*-11.2*/**-9.6** |
| Summer | -1.5/*-1.2*/*-0.8* | -4.0/**-6.1**/**-6.6** | -3.4/*-3.0*/**-2.0** |
| Autumn | -1.5/*-2.0*/**-2.0** | -10.1/*-10.4*/*-11.2* | -9.9/**-11.1**/**-4.6** |

**Table 2: Global average improvement in RF (W m$^{-2}$) at specified intervals after injection as compared to CESM2(WACCM6) control runs for FI (black), DI (orange), and ADI (green) injection schemes. Bold values indicate a statistically significant difference in mean RF between FI and the corresponding DBS-informed injection on that day. Gray shaded cells indicate times at which FI resulted in stronger RF than both DI and ADI.**

---

## Author Response (AR2)

Response to reviewer

Review:
The authors have appropriately addressed most of my comments, but there are still questions and concerns about the paper that require substantial revision of the text.

**First point: It seems like there is still confusion about injecting aerosols vs aerosol precursors and there needs more clarification and discussion throughout the text. The text often refers to aerosol injections, which is however not applied in most model studies and potentially for applications. Injecting aerosol precursors (not aerosols) will result in a different evolution and growth of aerosols, than aerosols. This is because the time it takes for nucleation of sulfur (SO2), as for example discussed in Mills et al. (2017). The lifetime of SO2 is usually about 30 days, but it depends on the availability of OH. After a large volcanic eruption, or large SO2 injections, OH depletion can lead to a prolonged lifetime of SO2 (47 days after MT Pinatubo). The application of the dispersion model using a tracer or aerosol is therefore different than injecting SO2. To outline the complexity, it has been shown that SO2 injections at a point location results in smaller aerosols than injections at a longitudinal band. This is because, the zonal wind is already dispersing the gas quickly and therefore reduce the amount at the injection location. More nucleation is induced instead of condensation on existing participles. Furthermore, after enhanced aerosol burden has been established in the stratosphere, sulfur injections will condensate on existing particles independent on the dispersion efforts.**

1) We thank the reviewer for pointing out that we need to be careful in our references to aerosols versus aerosol precursors; we have clarified this distinction throughout that none of our simulations are injecting aerosols but either consider passive tracer (referred to as neutral tracers or pseudo-aerosols) or an aerosol precursor. While we acknowledge the reviewer's point about differences in chemistry and aerosol microphysics, it is possible that there is some confusion about our methods. The purpose of the DBS-informed injection is to identify transport barriers and transport-enhancing features. While we identified these features using pseudo-tracers (effectively infinitesimal radiatively inert aerosols), identification of the barriers does not depend on the form of the tracers. The transport barriers will affect aerosols, gases, or anything else that is advected by the wind fields, and our second set of simulations indeed inject the precursor gas SO2 and not aerosols.
This has now been stated more clearly in lines 144-149:

*"We note that although Run #1 involves calculation of neutral tracers (resembling infinitesimal radiatively inert aerosols), Run #2 involves injection of the gaseous aerosol precursor SO$_2$. SO$_2$ requires time to convert to sulfate aerosols (e.g., Mills et al., 2017), and the injection strategy of SO$_2$ (for example along a longitudinal band instead of into a single grid box) has been demonstrated to affect aerosol size and hence radiative effects of the injection (e.g., English et al., 2012). Nevertheless, the purpose of these DBS-informed simulations is to describe the effects of recognizing transport barriers or atmospheric features that enhance transport. The applicability of this method is not dependent on whether a gas or particle is injected."*

**Second point: From what is shown in the paper, I am not convinced that this method leads to significant improvements. As shown in the paper, after a one-year simulation of the fully coupled model, it seems that there are no significant differences in coverage between the fixed injection method and the dynamically derived injection method in terms of efficiency (Fig 9). Fig 9 is showing a strong reduction in coverage in the first 30 days, which may have to do with the lifetime of SO2, and how long it takes to build up a larger sulfate coverage. It would be helpful to also show the absolute values of sulfate here, which are likely to be very small initially. Furthermore, Table 2 indicates, that after 30 and 365 days, in 2 out of 4 cases, the radiative forcing is more strongly reduced in the fixed injection case and the third case shows almost no difference between the DI method. I agree that this method may be useful to consider for the onset of sulfur injections and should be explored in more detail. However, I don't think the authors can support that there are "long-term gains in sulfate burden and radiation" as stated in the conclusions. I would therefore at this point advise against recommending this method as "a benchmark improvement in injection protocol".**

2)  We appreciate the reviewer's point about overstating some of our conclusions.  We have removed the two statements identified by the reviewer and have gone through the manuscript to ensure that our conclusions are supported by our results. Other statements have also been modified to most accurately reflect our findings.

As it stands, we have a rigorous, mathematically-supported theory for enhancing gas or aerosol dispersion along atmospheric structures. This theory is supported by our initial neutral-tracer experiments in simplified climate model wind fields. The influence of these injection choices is also present in our Run #2 simulations, though the intended effect is harder to predict. We do not feel that this negates our efforts, rather indicates that dynamic methods of injection location optimization should be further investigated.

Related to point #1, we reiterate that transport barriers apply to anything advected by the wind fields, be it aerosols or gaseous precursors.

**Finally, as stated in my earlier review, two co-authors of this study work on a feedback controller to improve climate impacts. It is important to address the question whether this method is suitable for applying a feedback-controller. Injection locations and amount have been chosen to improve the climate outcomes in particular of surface temperature. How could this method be integrated?**

3)  We have now added a short description of how a feedback algorithm might be used with DBS-informed injection in L485-491:

*"To manage uncertainties in atmospheric flow and climate response, several recent geoengineering climate modelling studies have employed a feedback algorithm that regularly adjusts the $SO_2$ injection rate (Jarvis and Leedal, 2012; MacMartin et al., 2014; Kravitz et al., 2017).  Future studies involving DBS-informed injection could pursue something similar. Outlining this process, every week of simulation, new injection locations would be determined*

*based on wind fields from the previous week, using the DBS algorithm described previously. The model would then be run forward for a week with the $SO_2$ injected at those locations. This process, which essentially constitutes a form of Model Predictive Control (Garcia et al., 1989), could be carried out for the length of the simulation."*

---

## Author Response (AR3)

Review:

**The authors have followed my suggestions regarding the first and third comment, but there are still some concerns regarding the second point. The authors state in their response that "the intended effect is harder to predict" in using the full Earth System model. As outlined in the earlier review, Table 2 does not show the expected improvement between the FI and the other cases, and at least half of the times the RF for the FI (fixed injection) case is larger than the other cases. As can be seen in the table for 365days, for winter and summer, the FI is showing a larger forcing (significantly larger in winter), while for spring and summer the RF is slightly smaller compared to the FI case (see Table 2 below). This should be mentioned in the text. One reason for this signal may be more complicated aerosol microphysics in the atmosphere, using SO2 injections than what can be estimated from a simple model. The method may be useful but has to be further investigated before recommending as an application for aerosol geoengineering.**

**The authors have added some minor changes to the wording in the Discussion and Conclusions. However, the abstract is still presenting results that are not reflected in the paper and therefore need to be changed:** "*Additionally, this enhanced dispersion slows aerosol microphysical growth, reducing the effective radii of aerosols at monthly timescales. This has long term impacts on radiative forcing, beyond the lifespan of the original influential transport barriers. We conclude that previous feasibility studies of geoengineering likely underestimate the potential cooling efficiency of sulfate aerosol geoengineering by not strategically injecting at optimized dispersion locations.*"

1. **The authors have not shown that the changes (increase and reduction, and not only reduction) in the effective radii of aerosols (at one theta level that is mostly above the enhanced aerosol layer in high latitudes) would change the radiative forcing. For example, it seems like the effective radius at 540K in spring (in Fig12) is largest for the simple case, but the radiative change (Table 2) does not reflect that and has a reduced radiative effect compared to the other cases.**
2. **The conclusion that the potential cooling efficiency of sulfate aerosol geoengineering is likely underestimated has not been shown in the paper.**

**The abstract would be much improved if the authors would point to the potential for this method, but also mention the need for further investigation of these methods in order to gain a more robust understanding.**

Thank you for your response to our previous changes. We have included changes to the abstract as well. The quoted section you highlighted now reads: *"Additionally, this enhanced dispersion slows aerosol microphysical growth, and can reduce the effective radii of aerosols up to 200-300 days after injection. While the long-term dynamics of aerosol dispersion are accurately predicted by short forecast transport barriers, the long-term influence on radiative forcing is more difficult to predict and warrants deeper investigation. Statistically significant changes in radiative forcing at timescales beyond the forecasting window showed mixed results, potentially increasing or decreasing forcing after one year when compared to fixed injections. We conclude*

*that future feasibility studies of geoengineering should consider the cooling benefits possible by strategically injecting sulfate aerosols at optimized time-varying locations. Our method of utilizing time-varying attracting and repelling structures shows great promise for identifying optimal dispersion locations and radiative forcing impacts can be improved by considering additional meteorological variables."*

---

## Author Response (AR4)

**Dear authors,**

**thank you for the changes in the abstract; I think your paper should be now published.**

**My recommendation is that you rethink the term "short forecast transport barriers", which has been
now introduced in the abstract. This term is perhaps not straightforward to understand for the general reader.**

**Congratulations and all the best**

**Rolf Müller**

Thank you for your suggestion. We have changed the respective sentence in the abstract to read as

*"While the long-term dynamics of aerosol dispersion are accurately predicted with transport barriers calculated from short forecasts, the long-term influence on radiative forcing is more difficult to predict and warrants deeper investigation."*